# Plant Extracts as Green Corrosion Inhibitors for Different Metal Surfaces and Corrosive Media: A Review

**Alan Miralrio** [1,*]  **and Araceli Espinoza Vázquez** [2,*]

1 Tecnologico de Monterrey, Escuela de Ingeniería y Ciencias, Ave. Eugenio Garza Sada 2501, Monterrey 64849, N.L., Mexico

2 Instituto de Investigaciones en Materiales, Universidad Nacional Autónoma de México, Circuito Exterior S/N, Cd. Universitaria, Coyoacán, Mexico City 04360, Mexico

\* Correspondence: miralrio@tec.mx (A.M.); arasv_21@yahoo.com.mx (A.E.V.)

**Abstract:** Natural extracts have been widely used to protect metal materials from corrosion. The efficiency of these extracts as corrosion inhibitors is commonly evaluated through electrochemical tests, which include techniques such as potentiodynamic polarization, electrochemical impedance spectroscopy, and weight loss measurement. The inhibition efficiency of different extract concentrations is a valuable indicator to obtain a clear outlook to choose an extract for a particular purpose. A complementary vision of the effectiveness of green extracts to inhibit the corrosion of metals is obtained by means of surface characterizations; atomic force microscopy, scanning electron microscopy, and X-ray photoelectron spectroscopy analysis are experimental techniques widely used for this purpose. Moreover, theoretical studies are usually addressed to elucidate the nature of the corrosion inhibitor—metal surface interactions. In addition, calculations have been employed to predict how other organic substances behave on metal surfaces and to provide experimental work with fresh proposals. This work reports a broad overview of the current state of the art research on the study of new extracts as corrosion inhibitors on metal surfaces in corrosive media. Most constituents obtained from plant extracts are adsorbed on the metal, following the Langmuir adsorption model. Electron-rich regions and heteroatoms have been found to be responsible for chemisorption on the metal surface, whereas physisorption is due to the polar regions of the inhibitor molecules. The plant extracts compiled in this work obtained corrosion inhibition efficiencies above 60%, most of them around 80–90%. The effect of concentration, extraction solvent, temperature, and immersion time were studied as well. Additional studies regarding plant extracts as corrosion inhibitors on metals are needed to produce solutions for industrial purposes.

**Keywords:** steel; copper; aluminum; green corrosion inhibition; plant extracts

## 1. Introduction

Metals are widely used in human activities due to their excellent mechanical and electrical properties [1–4]. In order to preserve the desired state of these metals, their preventive maintenance is a priority. Corrosion is probably the most common undesired phenomenon that leads metals to become weaker [5,6]. This natural process originates from the electrochemical interaction of metals with the corrosive environment. Sulfides, oxides, and others are generated through reactions between the metal surface and the corrosive medium [6–9].

Among metals, mild steel is the most widely used in the oil, food, energy, chemical, and construction industries due to its different applications, most of which are based on its excellent mechanical properties. This metal shows high mechanical resistance, durability, and toughness, among others,

which makes it a highly available material and at a relatively low cost. Consequently, solutions to problems related to the degradation by the corrosion of steel, mostly mild steel, is a high-priority topic. To a lesser extent, copper and aluminum alloys are studied as well. The high cost associated with corrosion, due to the replacement of rusted metals, can be reduced by using corrosion inhibitors [10].

### 1.1. Corrosion Inhibition Fundamentals

Constituting the archetypal source for mineralization and corrosion phenomena, seawater is an example of a common corrosive medium, containing an abundance of chloride ions, including massive waterbodies around the world: oceans, seas, and salt lakes [11]. On the other hand, hydrochloric acid (HCl) is frequently used both in decalcification processes and to produce corrosion under controlled conditions [1]. Several authors employ 1M HCl solutions, making it the most prominent corrosion medium to study corrosion since it is extremely aggressive and can be used to obtain an idea regarding corrosion on a certain metal [12–16]. Other solutions recurrently used as corrosive media are 3.5% NaCl [17–22], 0.1 M HCl [23], and 0.1 M NaOH [24–26]. To a lesser extent, 1M $H_3PO_4$ and 0.5 M $H_2SO_4$ are used as test solutions as well [27–31]

Corrosion inhibitors are substances that are added in small amounts on metal surfaces or are added to the corrosive medium, reducing the tendency to be affected by corrosion. The use of common corrosion inhibitors is sometimes limited, since these are based on dangerous substances for human health, such as chromium-based treatments [32]. Recent approaches take advantage of organic compounds that can be obtained from expired pharmaceutic drugs, mushroom extracts, and even plant extracts [33–37]. There is a variety of green organic compounds that function as corrosion inhibitors that show excellent properties in protecting metal surfaces, for example, derivatives of chitosan [38], phenylmethanimine [39], imidazoline [40], and ionic liquid [41]. In consequence, these compounds replace the traditional toxic corrosion inhibitors. Highly efficient corrosion inhibitors have been achieved by means of these substances, providing new recycling and reusing routes for drugs as well as corrosion inhibitors obtained from sustainable, ecological, and environmentally friendly sources, with plant extracts being a prominent group [25,42]. These extracts constitute another option of great interest, since they offer the possibility of a first approach to determine the class of natural compounds that help inhibit the corrosion process. The advantage is that making an extract from any plant is regarded as an uncomplicated task, thus allowing more efficiency at both extraction and use of these substances for experimentation.

An extract is a solution composed by the active principles of a plant or its parts (Figure 1) and a certain medium acting as solvent. The extraction yields depend on the polarity of the solvent used, in the techniques or methods (Soxhlet and maceration), among others. Active principles contained in the extract give the properties for a particular purpose. Thus, a given plant, in terms of its active principles and concentrations, can be associated with some benefits. These extracts are mostly known by their antioxidant, anti-inflammatory, antiviral, or antimicrobial effects. In addition, their corrosion inhibitor properties can be considered as synergic effects. Extracts are commonly obtained from the whole plant or the parts containing higher concentrations of active principles, named phytochemicals [43,44]. According to the literature, extracts of plants, fruits, seeds, flowers, and leaves contain active compounds that are promising for corrosion inhibition in aggressive media. Moreover, these compounds become cheap, widely available, and renewable corrosion inhibitor alternatives [45–49]. Thus, a review on the novel plant extracts that have been proven to be highly efficient corrosion inhibitors is necessary. This manuscript provides a wide landscape of the recently reported plant extracts as corrosion inhibitors in steel as well as aluminum and copper alloys. Basic aspects of the extraction methods, characterization techniques, theoretical modelling, and adsorption mechanisms are also discussed.

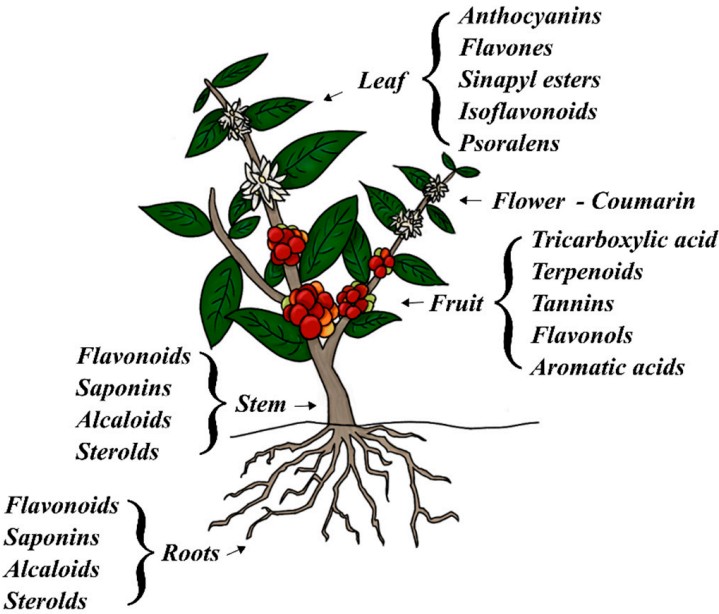

**Figure 1.** Basic parts of a plant and their common active compounds.

*1.2. Extraction Methods*

The vast variety of extraction methods used nowadays exceed by far the purpose of this review. However, a brief review on extraction methods is included below. The first step is to choose the part of the plant with the major concentration of active compounds of interest. All plant parts—leaves, flowers, seeds, fruits, roots, and stems—are used to obtain the extracts. In brief, extraction methods are based on heating, cooling, and separating the active compounds in the presence of the solvent.

Secondly, traditional extraction methods can be summarized in maceration, infusion decoction, digestion, and percolation (Figure 2) [50]. In general, the form of the extract method is applicable on the basis of what is desired to be obtained. In maceration, crushed, smashed, or cut materials, sometimes previously dried, are immersed in the extraction solvent for periods of at least 3 days in constant agitation (Figure 2a). The diffusion of the solvent in the targeted material solubilizes the active compounds, leading to their possible extraction. The solids suspended in the resulting mixture can be separated by filtering. For this method, the advantage is that all the essence is extracted without altering it, and the active ingredients are easily soluble. The infusion method yields the extract by means of maceration for a short period of time in the presence of boiling water. Thus, the most soluble constituents are solubilized, passing to the extract. Similarly, in the decoction process, the crude drug is boiled in a specific volume of water for a defined time (Figure 2c). The digestion method proposes the maceration of the raw materials in the presence of a slightly warm solvent, improving the solubility of the extraction solvent and preserving the active compounds from decomposition (Figure 2). Percolation is a filtration method, at room temperature, in which the moistened raw material is placed in a conical vessel, the percolator, with an adjustable closure (Figure 2b). Then, the percolator must be filled with solvent and covered up, obtaining the extract drop by drop [51]. The advantages of percolation lie in the high performance of active substances, in the short time required for their manufacture and the economy of the materials used.

More sophisticated methods are hot continuous extraction and ultrasound extraction or sonication [52]. The first one uses the Soxhlet apparatus, formed by a glass body with boiling flask, a siphon arm, thimble, extraction chamber, and condenser (Figure 2d) [52]. In brief, the boiling flask containing the solvent is heated and the vapor produced is condensed. The resultant liquid falls into the thimble containing the raw material, and the extract fills the extraction chamber up in order to put into function the siphon arm to return the liquid into the boiling flask. The reflux process must be stopped up to obtain the degree of extraction desired. Finally, sonication is a technique that uses

high energy ultrasounds to improve permeability of cell walls, producing cavitation to disrupt cellular membranes (Figure 2e) [53]. Consequently, sonication breaks the cells, releasing their content for further extraction. Liquids obtained by the methods introduced above are then clarified by filtration or decantation.

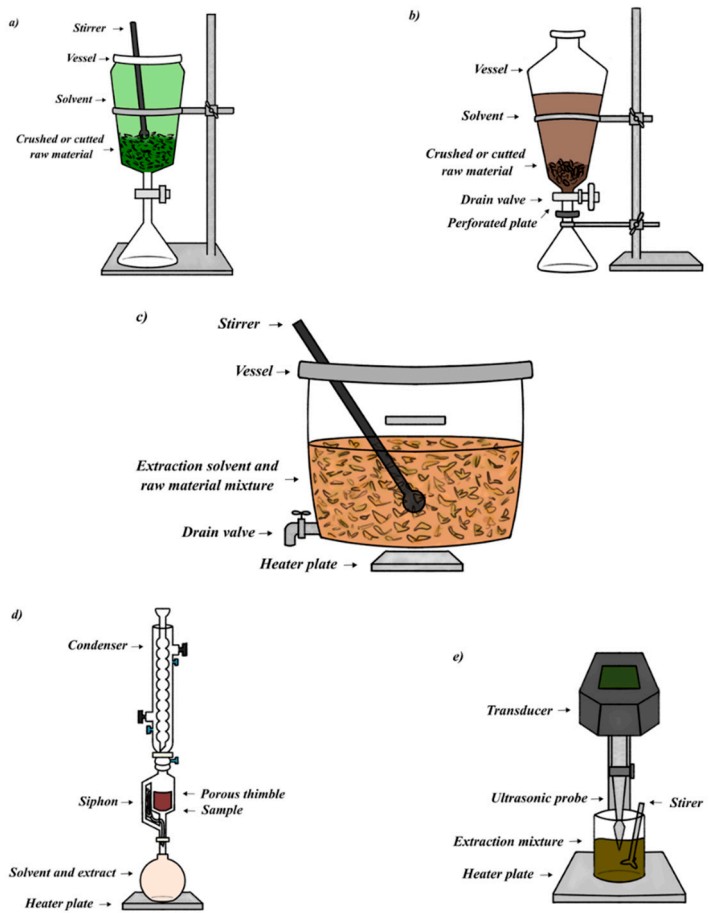

**Figure 2.** Diagrams of common extraction methods: (**a**) maceration, (**b**) percolation, (**c**) decoction, (**d**) Soxhlet, and (**e**) sonication.

The solvent has a key role in the extraction methods, since it is responsible for solubilizing the active compounds when it diffuses through plant tissues, making their extraction possible. Extraction solvents have shown to affect the physical, chemical, and antioxidant properties of the extracts obtained, as the concentration of flavonoids, saponins, phenolic compounds, and others present in plant extracts vary according to the extraction solvent [54–56]. Consequently, various solvents have been used to obtain the desired concentration of active compounds from plant extracts. Then, the efficient extraction of active compounds depends on the solvent used, among which the most common are water, methanol, ethyl acetate, dichloromethane, and hexane [54–56]. Water could be the most convenient extraction solvent, since it is highly available, non-toxic, non-flammable, and inexpensive [57–59]. Not all plant extracts can be obtained as aqueous extracts, giving the chance to test other solvents. Thus, solvents are selective and, in order to obtain the optimal yield, several options must be tested.

Other important parameters for the extraction process are drying and extraction temperatures. The first one marks the temperature to get the plant dry, although room temperature is commonly used since plants are usually dried in the shade. The use of dried plants has benefits, being suitable for long storage and with reduced weight. In contrast, the drying process has been shown to affect the stability of bioactive compounds and the antioxidant capacity [54,60]. Fresh plants can be used instead of dried ones, although their use could produce some drawbacks as well. Fresh plants could be subject to degradation by solar radiation, and some of their constituents could be easily evaporated

or oxidized. However, in some cases, phytochemicals could be extracted in higher concentrations from fresh samples instead of dried ones [61–63]. The extraction temperature is another important parameter since a high temperature promotes the decomposition of phytochemicals and a lower one reduces the solubility of active compounds and hinders its extraction. To achieve the optimal concentration of phytochemicals requires the correct choice of extraction temperatures, extraction cost, solvent, and others [64].

### 1.3. Characterization Techniques

A proper characterization of the extracts proposed as corrosion inhibitors is needed; consequently, several experimental techniques are available for this purpose. Cyclic potentiodynamic polarization (PP) is used for evaluating the susceptibility of a metal to localized corrosion such as pitting and crevice [65]. Polarization tests, such as PP, are based on the evaluation and analysis of the current produced by a variable potential in a working electrode [65]. Another recurrently used technique to study anti-corrosion performance, in considerable short testing times, is electrochemical impedance spectroscopy (EIS) [66]. This technique is used to determine the impedance of a system in terms of the frequency of a variable potential. The analysis of EIS results relies on models with equivalent electrical circuits, with the most recurrent graphical representations of its results being Nyquist plots [67]. EIS shows more information, for example, mechanism and different resistance of the system. Linear polarization resistance (LPR) is a technique used to obtain the corrosion rate by determining the relationship between electrochemical potential and generated currents on charged electrodes [68]. Less sophisticated and time-consuming is the weight loss method (WL), since this technique is based on the mass lost by corrosion, which is directly monitored to get the corrosion rate. Some properties measured in the presence and absence of the substance used as corrosion inhibitor, such as corrosion current density obtained by PP, can be used to obtain the inhibition efficiency [66].

Surface characterization is commonly addressed by means of spectroscopy and microscopy techniques. Scanning electron microscope (SEM) provides a clear comparison between the metal surface with and without a corrosion inhibitor, as well as other morphological information [69,70]. Similarly, the atomic force microscope (AFM) obtains information regarding the shape of the metal surface for comparison purposes and topography imaging [71–74]. X-ray photoelectron spectroscopy (XPS) is recurrently used for oxidation states, stoichiometry, and electronic state determination [75–78]. Complementary characterizations are usually done through Fourier transform infrared spectroscopy (FT-IR) to obtain information on the functional groups and vibrational modes on the corrosion inhibitors. Similarly, ultraviolet–visible spectroscopy (UV–VIS) helps to elucidate functional groups, electronic transitions, and optical band gaps.

Corrosion inhibitors can be classified in the manner they inhibit corrosion: cathodic, anodic, or mixed-type inhibitors [79–81]. Cathodic corrosion inhibitors decrease the corrosion potential, towards lower values, inhibiting the reactions that take place at the cathode, such as oxygen reduction and hydrogen evolution. Anodic corrosion inhibitors move the corrosion potential towards higher values and interact with the reactive sites on the metal surface, passivating them. Mixed-type inhibitors are those that cannot be classified as cathodic or anodic [79–81]. These inhibitors can protect the metal surface in three possible manners: physisorption, chemisorption, and film formation. Physisorption is motivated by the electrostatic interaction between inhibitor molecules and the metal surface, whereas chemisorption is due to donor–acceptor interactions between vacant orbitals on the metal surface and free electron pairs in the inhibitor [79–81]. Lastly, film formation provides the metal surface with a physical barrier from the corrosive media, protecting it from corrosive attacks.

### 1.4. Adsorption Mechanism and Quantum Chemistry Methods

Phytochemicals present in plant extracts are used as corrosion inhibitors since the inhibition effect is due to the adsorption of the inhibitor molecules on the metal surface, providing the metal with a protective film by blocking the active sites. Consequently, this coverage is less reactive than the blank surface. Several organic compounds coming from natural sources, such as (-)-epicatechin gallate, catechin, or (-)-epicatechin and vallinin, have been proposed since their compounds exhibit some properties argued as requirements to act as corrosion inhibitors. Heteroatoms, substituted aromatic rings, and $\pi$ electrons are claimed to promote the adsorption of the corrosion inhibitor molecules [82–86]. Moreover, some polar functional groups make the inhibitor more soluble in common corrosive media and tend to act as the adsorption sites of the corrosion inhibitor molecule.

Characterization of corrosion inhibitor adsorption on the metal surface by means of Gibbs adsorption energy $\Delta G^{\circ}_{ads}$ is sometimes complemented by theoretical studies. Accordingly, several theoretical frameworks have been used through the years [87–92]. Certainly, the density functional theory (DFT) is the most recurrently used method since it provides reliable results in affordable computational time. The simplest approach considers the molecules of the most relevant constituents. From the optimized structures of these molecules, several structural, electronic, thermodynamic, and chemical properties can be obtained. $E_{HOMO}$, $E_{LUMO}$, energy gap, electronegativity, global hardness, Fukui reactivity indices, and fraction of electron transferred from the corrosion inhibitor molecule to the targeted metal are a group of quantum chemical parameters commonly obtained through methods within DFT.

More robust theoretical studies include the adsorption of corrosion inhibitor molecules in the targeted metal. Two approaches are usually adopted: periodic surfaces [14,93–95] and clusters [96–103]. Free binding energy $G_{bind}$ has been proven to be comparable with Gibbs adsorption energy $\Delta G^{\circ}_{ads}$ experimentally obtained. Calculations on DFT are complemented with solvent models, implicit and explicit, and corrections to total energy due to dispersion interactions. Along with DFT, molecular dynamics (MD) is another theoretical framework employed to study the adsorption of corrosion inhibitor molecules. With this method, it is possible to study explicit interactions among corrosion inhibitor molecules, corrosion medium molecules, and the metal surface. In order to obtain a global overview of the adsorption mechanism, thermodynamic variables such as temperature and number of molecules can be studied as well.

To provide a broad outlook on the extracts that have been recently evaluated to act as corrosion inhibitors, the following sections detail information regarding these substances. Extracts, showing good corrosion inhibition properties, are divided in terms of the targeted metal.

## 2. Plant-Based Corrosion Inhibitors for Mild Steel

Steel is the most used metal to test corrosion inhibition due to its innumerable applications. The current section deepens on the most recent reports of plant extracts evaluated for their use as corrosion inhibitors on steel.

Table 1 shows the equations to determine the inhibition efficiency (IE) by different techniques.

**Table 1.** Equation to determine the inhibition efficiency.

| Technique | Equation |
|:---:|:---:|
| **EIS** [104] | $IE\ (\%) = (1 - ((R_p)_{blank}/(R_p)_{inhibitor}) * 100$ |
| **PP** [105] | $IE(\%) = \left[1 - \dfrac{icorr}{icorrBlank}\right] \times 100$ |
| **WL** [106] | $IE\ (\%) = ((W_0 - W_1)/(W_0)) * 100$ |

$R_p$ is a polarization resistance—with inhibitor, it is $(R_p)_{inhibitor}$, and without it, it is $(R_p)_{blank}$; $i_{corr}$ is the current density with and without inhibitor; and $W_0$ and $W_i$ are the weight loss values in absence and presence of the inhibitor, respectively.

In order to complement the description on the corrosion inhibition provided by the techniques introduced above, several authors include the study of the adsorption of corrosion inhibitor molecules on the metal surface [96,107–109]. Several adsorption isotherm models have been proposed to describe the adsorption mechanism of organic inhibitors on the metal surface. The most appropriated model is the one that best fits the experimental values. The following table (Table 2) summarizes the most used adsorption isotherm models. In Table 2, $C_R$ is the concentration of the inhibitor, θ is the degree or surface coverage of inhibitor, and $K_{ads}$ is adsorption equilibrium constant. In most cases [2,110–113], the model that best fits to extracts as corrosion inhibitors on metals is the Langmuir isotherm. Adsorption equilibrium constant $K_{ads}$ is directly related to Gibbs adsorption energy $\Delta G°_{ads}$ by the equation

$$\Delta G°_{ads} = -55.5 \, RTlnK_{ads} \tag{1}$$

where $R$ is the universal gas constant, $T$ is the absolute temperature, $K_{ads}$ is the adsorption equilibrium constant, and 55.5 is the water solution concentration in mol/L units. The negative sign of $\Delta G°_{ads}$ denotes the spontaneous adsorption of the corrosion inhibitor molecules on the metal surface. Values below −40 kJ/mol are related to chemisorption, ranging from −40 to −20 kJ/mol to mixed physisorption–chemisorption regime and above −20 kJ/mol to physisorption [114–120].

**Table 2.** Main adsorption isotherm models.

| Model | Adsorption Isotherm | Reference |
|:---:|:---:|:---:|
| langmuir | $\frac{C_R}{\theta} = \frac{1}{K_{ads}} + C_R$ | [121,122] |
| temkin | $\theta = lnC_R + K_{ads}$ | [121,123,124] |
| freundlish | $log\theta = nlogC_R + logK_{ads}$ | [124,125] |
| flory–huggins | $log\frac{\theta}{C_R} = blog(1-\theta) + logK_{ads}$ | [125] |
| frumkin | $logC_R\frac{\theta}{1-\theta} = 2\alpha\theta + 2.303logK_{ads}$ | [121] |
| el-awady | $log\frac{\theta}{1-\theta} = ylogC_R + logK, K_{ads} = K^{1/y}$ | [121,126] |

Table 3 summarizes the maximum corrosion inhibition efficiency achieved with several plant extracts. It is pertinent to mention that, under testing conditions, plant extracts listed in Table 3 exhibited relevant corrosion inhibition efficiencies, ranging from 64.24 to 98.8%, reported for *Ipomoea batatas* extract [127] and *Glycyrrhiza glabra* (Persian liquorice) extract [128], respectively. Moreover, Table 3 includes the solvent used to obtain the plant extracts, metal and corrosive medium tested, as well as the experimental characterizations done. As previously discussed, mild steel is the most studied among iron-based alloys (Table 3) and 1 M HCl is the most used corrosive medium to test corrosion inhibitor (Table 3).

As discussed previously, all parts of a plant can be used to obtain extracts. For instance, Al Hasan and coworkers studied stem aqueous extract of *Bacopa monnieri* and *Lawsonia inermis* (henna) on low carbon steel in a solution of 0.5 M NaOH [25]. The extract was obtained from 10 g powder of *Bacopa monnieri* and 5 g of henna and tested at inhibitor concentrations of 0.5, 1, and 2%. The joint work of this stem extract produced a mixed type inhibitor of considerable corrosion inhibition efficiency. The authors found a maximum inhibition efficiency about 80% by the PP technique. Moreover, weight loss measurements resulted in an inhibition efficiency of 65% (Table 3).

**Table 3.** Basic information and parameters obtained with several plant extracts evaluated as corrosion inhibitors in steel: plant, extraction solvent, metal, and corrosive medium used for corrosion inhibition performance tests; temperature and concentration C test ranges; maximum inhibition efficiency obtained ($\eta_{max}$); and characterizations reported are included.

| Plant | Extraction Solvent | Metal | Corrosive Medium | Temperature (K) | C | Characterization | $\eta_{max}$ (%) | Reference |
|---|---|---|---|---|---|---|---|---|
| *ananas comosus* | Water | Low-carbon steel | 1 M HCl | 308–338 | 1000 ppm | WL, EIS, PP, UV–VIS, SEM | 97.6 | [136] |
| *artemisia herba-alba* | Water | Stainless steel | 1 M $H_3PO_4$ | 298–353 | 1 g/L | EIS, SEM-EDS, GC–MS | 88 | [137] |
| *bacopa monnieri/lawsonia inermis* | Water | Low-carbon steel | 0.5 M NaOH | - | 0.4–0.8 g/L | WL, PP | 80 | [25] |
| *camellia sinensis* | Water | Carbon steel | 3.5% NaCl | - | 0.5–2% | LPR, LC–MS | 80 | [82] |
| *cryptocarya nigra* | Hexane, dichloromethane, methanol | Mild steel | 1 M HCl | - | 10–1000 mg/L | PP, EIS, SEM-EDS, FT-IR, UV–VIS | 91 | [135] |
| *dioscorea septemloba* | Ethanol | Carbon steel | 1 M HCl | 303.15 | 0.1–2.0 g/L | NMR, EIS, FT-IR, SEM | 72 | [138] |
| *eucalyptus* | Water | Mild steel | 1 M HCl | - | 800 ppm | PP, WL, EIS, FT-IR, UV–VIS, SEM, AFM | 88 | [127] |
| *eucalyptus globulus* | Water | Low-carbon steel | 0.5 M $H_2SO_4$ | - | 100–600 mg/L | WL, EIS, AFM, FT-IR, SEM, EFM | 93.09 | [129] |
| *euphorbia heterophylla llinneo* | Water | Mild steel | 1.5 M HCl | - | 1 g/L | PP | 69 | [139] |
| *ficus tikoua* | Water | Carbon steel | 1 M HCl | - | 200 mg/L | FT-IR, EIS, PP, SEM | 95.8 | [130] |
| *glycyrrhiza glabra* | Water | Mild steel | 1 M HCl | - | 200–600 ppm | EIS, PP, FT-IR, AFM | 88 | [128] |
| *ipomoea batatas* | *N*-hexane | Galvanized steel | 1 M HCl | 308.15–338.15 | 0.7 g/L | FT-IR, EIS | 64.26 | [140] |
| *juglans regia* | Water | Mild steel | 3.5 wt % NaCl | - | 1000 ppm | PP, EIS, FR-IR, SEM | 94.2 | [141] |
| *luffa cylindrica* | Ethanol | Mild steel | 1 M HCl | 303–333 | 1 g/L | WL, PP, EIS, FT-IR | 87.98 | [142] |
| *nicotiana tabacum* | Water | Q235 steel | 0.1 M NaOH | - | 34 mg/L | PP, FT-IR, XPS, SEM, XRD | 83.9 | [143] |
| *olea europaea* | Methanol, ethyl acetate, dichloromethane, hexane | Mild steel | 0.1 M NaOH + 0.5 M NaCl | - | 200–800 mg/L | EIS, GC–MS | 91.9 | [134] |
| *papaver somniferum* | Ethanol | AISI 304 stainless steel | 0.2 M HCl | 298.15–318.15 | 100–500 ppm | AFM, SEM-EDS | 88 | [144] |
| *pterocarpus santalinoides* | Water, ethanol, and methanol | Low carbon steel | 1 mol/dm$^3$ HCl | 298.15–333.15 | 0.7 g/L | EIS, PP, LPR, SEM-EDAX, AFM | 90 | [145] |
| *rosa canina* | Water | Mild steel | 1 M HCl | - | 200–800 ppm | EIS, UV–VIS, PP, SEM | 86 | [146] |
| *saraca ashoka* | Water | Mild steel | 0.5 M $H_2SO_4$ | - | 25–100 mg/L | WL, EIS, PP, SEM, UV–VIS, FT-IR, AFM | 93.09 | [131] |
| *tamarindus indiaca* | Water | Mild steel | 3.5% NaCl | - | 300–1000 ppm | FE-SEM, AFM, EIS, FT-IR, GI-XRD | 96 | [147] |
| *tamarindus indiaca* | Water | Mild steel | 3.5% NaCl | - | 1000 ppm | FT-IR, UV–VIS, XRD, TGA, Raman, EIS, FE-SEM | - | [132] |
| *taraxacum officinale* | Water/ethanol | Carbon steel | Seawater | 295.15–328.15 | 100–400 mg/L | WL, PP, EIS, SEM, FT-IR, UV–VIS | 94.3 | [133] |
| *tinospora crispa* | Water, acetone/water | Mild steel | 1 M HCl | - | 800–1000 ppm | WL, EIS, SEM | 80 | [111] |
| *tithonia diversifolia* | Water | Mild steel | 1 M HCl | - | 0.1–0.7% | WL, EIS, PP, FT-IR | 94.55 | [148] |
| *zingiber officinale* | Methanol | Mild steel | 1 M HCl | 298.15–328.15 | 100 ppm | PP, EIS, FR-IR, UV–VIS, AFM | 92.5 | [149] |
| *ziziphora* | Water | Mild steel | 1 M HCl | - | 200–800 ppm | FT-IR, UV–VIS, EIS | 93 | [150] |

Leaf extracts are also used to obtain extracts with corrosion inhibition properties. For instance, green *Eucalyptus* leaf extract as corrosion inhibitor, in mild steel in 1 M HCl corrosive medium, was evaluated by Dehghani, Bahlakeh, and Ramazanzadeh [127]. The maximum corrosion inhibition achieved with this aqueous extract is 88% at 800 ppm concentration (Table 3). EIS showed increment of charge transfer resistance as the extract concentration increased. On the basis of PP tests, the mixed nature of the corrosion inhibition of the *Eucalyptus* leaf extract was evidenced. SEM and AFM tests were used to demonstrate that higher concentrations increased the number of protective molecules adsorbed on the metal surface. Gibbs adsorption energy $\Delta G^{\circ}_{ads}$, calculated by fitting the Langmuir isotherm, showed that the inhibitor concentration increased the chemisorption component of this mixed type inhibitor.

The aqueous green tea (*Camellia sinensis*) extract was evaluated by Pradipta and coworkers as a corrosion inhibitor on carbon steel in a 3.5% NaCl corrosive medium [82]. Green tea was chosen as substances with multiple polar atoms and electron-rich bonds, such as the natural antioxidants contained in this plant, are potential mixed-type corrosion inhibitors. Through linear polarization resistance measurements, green tea extract exhibited a similar corrosion inhibition efficiency, ranging from 51 to 70%, in comparison of commercial calcium nitrite corrosion inhibitor at similar concentrations (Table 3). At equal volumes, the green tea extract exhibited a higher corrosion inhibition efficiency of about 75–80% in comparison of the commercial corrosion inhibitor. On the basis of liquid chromatography–mass spectrometry, (-)-epigallocatechin gallate, (-)-epicatechin gallate, and catechin or (-)-epicatechin are supposed to form the mixed-type corrosion inhibiting film.

The *Eucalyptus globulus* leaves extract has been tested by Haldhar and Prasad [129] on low carbon steel immersed in a 0.5 M sulfuric acid ($H_2SO_4$) solution. Corrosion inhibition properties were evaluated using weight loss and EIS (Table 3). This investigation shows that the Nyquist diagram reached a maximum value in the real part of the impedance of about 250 $\Omega cm^2$. This extract obtained its strongest corrosion inhibition of 93.09% at a concentration of 600 mg/L. Through UV–VIS spectroscopy, the formation of coordination bonds between inhibitor molecules and $Fe^{2+}$ was verified. SEM and AFM tests are used to confirm the formation of the corrosion protective film on the metal surface. Various functional groups containing heteroatoms and unsaturation in the phytochemical constituents of the plant extract are studied by using FT-IR and proton-nuclear magnetic resonance ($^1$H NMR) techniques, respectively.

Another aqueous extract tested to act as corrosion inhibitor on carbon steel in 1 M HCl corrosive solution is the *Ficus tikoua* leaf extract [130]. According to Wang and coworkers, PP curves obtained agree with a mixed-type inhibitor. The highest corrosion inhibition efficiency achieved, of about 95.8%, was obtained at 298 K with a 200 mg/L concentration (Table 3). Authors argued that the complex composition of the extract hinders fitting an adsorption isotherm to calculate Gibbs adsorption energy $\Delta G^{\circ}_{ads}$. Through SEM imaging, the surface smoothness gained by applying the extract in comparison of the blank surface was confirmed.

In addition, fruit extracts have been proposed as corrosion inhibitors. The corrosion inhibition performance of the *Rosa canina* fruit aqueous extract was evaluated on mild steel in a 1M HCl electrolyte by Sanaei and coworkers in 2019. The properties of the extract were evaluated by means of electrochemical impedance spectroscopy, UV–VIS spectroscopy, and SEM imaging. SEM images confirmed the lower degradation obtained by applying the *Rosa canina* fruit aqueous extract on mild steel. The maximum corrosion inhibition efficiency achieved, of about 86%, was obtained with an 800 ppm concentration (Table 3). Moreover, the extract behaved as a mixed-type inhibitor, with both cathodic and anodic reactions. It has been suggested that physically adsorbed molecules (extract) bind to metal in local cathodes and the dissolution of the metal is delayed when the cathodic reaction is prevented, while molecules chemically adsorbed protect the anodic areas.

As studied by Saxena and coworkers, *Saraca ashoka* seed aqueous extract on mild steel in 0.5 M $H_2SO_4$ behaves as a good corrosion inhibitor [131]. According to electrochemical impedance spectroscopy and weight loss measurements, a maximum corrosion inhibition efficiency of about 95.48

and 93.09%, respectively, was achieved with a 100 mg/L extract concentration (Table 3). The formation of a protective film on the metal surface was confirmed by AFM and SEM imaging. For this extract, the Langmuir adsorption model was a better fit. The study concluded that *Saraca ashoka* seed aqueous extract behaves as a mixed-type corrosion inhibitor.

Akbarzadeh and co-workers studied an intelligent self-healing anticorrosion coating using nanocarriers on the basis of a graphene oxide (GO) nanoplatform with *Tamarindus indiaca* extract (Ti.E) and $Zn^{2+}$ ions, labeled as GON-Ti.E-Zn, obtained through a green assisted route [132]. Raman spectroscopy results revealed the defect creation on the GO nanosheets due to the Ti.E molecule adsorption. The adsorption of Ti.E molecules on GO sheets was confirmed by several experimental techniques, including XRD, FT-IR, and UV–VIS. EIS spectra of the epoxy ester coating with and without artificial scratches along with salt spray observation demonstrated the achievement of an active barrier system, exploiting simultaneously *Tamarindus indiaca* extract and Zn-modified GO nanocarriers. In fact, the molecules released within GO, previously contained in Ti.E and zinc cations, are assumed to be the components responsible to build the protective layer. In this case, corrosion inhibition efficiency is not reported [132].

The aqueous extract of *Tinospora crispa* as corrosion inhibition on mild steel in 1 M HCl was evaluated by Hussin and co-workers [111]. Weight loss, EIS, and PP methods are used for deepening into the corrosion inhibitor properties of the extract. The maximum inhibition, of about 70–80%, was obtained with an 800 ppm concentration (Table 3). Another extract, obtained with acetone–water as extraction solvent, has been studied as well. In this case, the maximum inhibition efficiency was obtained with a 1000 ppm concentration. The Langmuir adsorption isotherm was used to obtain a Gibbs adsorption energy $\Delta G^{\circ}_{ads}$ of about −21.87 kJ/mol with the aqueous extract and −20.25 kJ/mol with acetone–water extract. The spontaneous physisorption process was determined by these calculations. SEM imaging confirmed the protection provided by the extracts on the mild steel surface.

Deyab and Guibal studied the *Taraxacum officinale* extract, obtained in a 2:3 aqueous/ethanol solution solvent, on carbon steel in seawater to test desalinization plant conditions [133]. This extract, according to the weight loss method, as well as PP and EIS tests, showed high inhibition efficiencies of up to 94.3% (Table 3). PP data showed that the extract behaved mostly as an anodic inhibitor. The inhibition mechanism was found through a modified Langmuir adsorption isotherm, as physisorption of the compounds contained in the extract. Moreover, Nyquist plots showed that the inhibition efficiency increased as the extract concentration increased. From all the above, authors proposed *Taraxacum officinale* extract as a potential corrosion inhibitor for cooling systems and desalination plants.

## 2.1. Solvent Effect

Since several solvents can be used to obtain extracts, the behavior of the inhibition efficiency of the different extract obtained is another topic of interest. This synergistic effect can be related to its higher effectiveness to extract the phytochemicals responsible for the inhibition effect. Thus, different solvents must be tested in order to determine the one that obtains the extract with the best performance [4]

The investigation of olive leaf (*Olea europaea*) extract performed by Ben et al. [134], tested on mild steel in a 0.1 M NaOH + 0.5 M NaCl corrosive solution, showed that inhibition efficiency increases as the polarity of the extraction solvent increases. They found that the inhibition efficiency decreased as follows: methanol, ethyl acetoacetate, hexane, and dichloromethane. The maximum inhibition efficiency was obtained with the methanol extract of about 91.9% (Table 3). On the basis of gas chromatography–mass spectrometry analysis, the authors suggested that the inhibition activity could be due to the presence of nitrogen, oxygen, and π-electrons. The extract was found to be a phenol- and flavonoid-rich substance. According to PP studies, olive leaf extract is a mixed-type inhibitor. EIS and Mott–Schottky analyses agreed well with PP results.

On the other hand, Faiz et al. used the *Cryptocarya nigra* extracts obtained with three different solvents (hexane, dichloromethane, and methanol) and three alkaloids (*N*-methylisococlaurine, *N*-methyllaurotetanine, and atherosperminine) isolated from the dichloromethane extract to act

as corrosion inhibitors on mild steel in 1 M HCl. These alkaloids were chosen by their antioxidant properties and their poly-phenolic groups, expected to assist the protection of the metal surface [135]. It is interesting that the *Cryptocarya nigra* dichloromethane extract is a potent corrosion inhibitor and *N*-methyllaurotetanine achieved the highest efficiency of about $\eta_{max\sim}$ 91.05 and 88.05%, respectively (Table 3). The alkaloid behaves as a good corrosion inhibitor since it contains nitrogen. Moreover, the authors argued that *N*-methyllaurotetanine has more oxygenated functional groups in comparison to the others and a rigid structure, being more favorable to protect the metal surface. According to Gibbs adsorption energy ($\Delta G^{\circ}_{ads}$) obtained by Langmuir adsorption isotherm plots, the most efficient extracts were adsorbed on the metal surface via physisorption [135].

The crude extract of *Pterocarpus santalinoides* leaves, studied by Ahanotu et al. [145], has been shown to be effective in inhibiting the corrosion in low carbon steel in a 1 mol/dm$^3$ HCl solution. Results obtained through EIS, PP, and LPR measurements showed that the protection efficiency of the *Pterocarpus santalinoides* leaf extract improves with an increase in dosage and temperature. The metal surface has been protected by over 90% at 333.15 K (Table 3). The authors suggest that the extract behaves as a mixed-type corrosion inhibitor according to its PP results. The lack of roughness in AFM measurements reveal that the surface is not deeply penetrated with the use of *Pterocarpus santalinoides* leaf extract, in contrast with the corrosion showed without using it. The performance of the extracts obtained with different extraction solvents follows the order ethanolic > methanolic > aqueous. The authors suggested that this tendency is a consequence of the better efficiency of ethanol and methanol to extract the flavonoids contained in *Pterocarpus santalinoides* leaves, compounds known to act as good inhibitors in carbon steel [4].

Akbarzadeh et al. [147] studied the performance of green corrosion inhibitor obtained from the *Tamarindus indiaca* (TAM) extract mixed with zinc nitrate (ZS). The metal tested was mild steel in 3.5% NaCl as corrosion medium. Electrochemical impedance spectroscopy results showed the synergistic behavior and 96% corrosion inhibition efficiency in TAM with 300 ppm and ZS 700 ppm after a 24 h immersion (Table 3). Polarization spectrum results exhibited the dominant behavior of the anodic depression in the mixture of TAM and ZS. Field emission SEM and grazing incidence XRD images confirmed the formation of a uniform protective layer.

## 2.2. Temperature and Immersion Time Effect

Temperature has an important influence on the phenomenon of corrosion in metal surfaces. It is possible to modify the interaction between the corrosive medium and the metal surface in the presence of the extract with inhibitors. Some extracts exhibit an increasing inhibition efficiency tendency towards higher temperatures [151]. However, other extracts show different behaviors. Thus, the evaluation of the inhibition efficiency as a function of the temperature is important since every extract could perform differently [152–157]. Similarly, the immersion time is another factor that could modify the inhibition efficiency, and consequently some authors have evaluated it as well [158–161].

For instance, Wang et al. evaluated the corrosion inhibition performance of the tobacco rob (*Nicotiana tabacum*) aqueous extract in Q235 steel in artificial seawater with a 0.1 M NaOH solution as corrosive medium [143]. Nicotine was found to be the main compound responsible of the corrosion inhibition effect. Inhibition efficiency increased as the extract concentration and temperature increased. The maximum corrosion inhibition performance, inhibition efficiency of about 83.9%, was obtained with a 100 mg/L extract concentration at 333.15 K (Table 3). According to XRD spectra and SEM images, without the extract, $CaCO_3$ and $CaSO_4$ deposit on the metal surface. In contrast, the extract retarded the growth of both deposits. Authors suggest that the growth of $CaCO_3$ and crystal $CaSO_4$ is blocked by chelating due to the water-soluble -OH group contained in the extract. XPS indicated that the corrosion inhibition was due to chemisorption on the metal surface.

As discussed previously, all parts of plants can be used to obtain extracts. Pineapple (*Ananas comosus*) stem extracts were evaluated by Mobin, Basik, and Aslam as a corrosion inhibitor on low-carbon steel immersed in 1 M HCl [136]. A high maximum inhibition efficiency of about 97.6% was obtained

with this extract at a 1000 ppm concentration and 338 K temperature (Table 3). A dependence on the electrolyte temperature and inhibitor concentration was observed, showing that both properties increase towards higher temperatures and concentrations, respectively. The adsorption was found, according to the Langmuir adsorption isotherm fitted, as mixed type physisorption–chemisorption. Higher temperature leads to chemisorption, in agreement with the tendency observed for inhibition efficiency. Results obtained through WL, EIS, and PP methods show consistency among them [136]. A smoother metal surface is obtained by the action of the inhibitor extract, as shown by SEM images.

The *Tithonia diversifolia* flower extract as a corrosion inhibitor on mild steel in 1 M HCl was tested through electrochemical impedance spectroscopy, weight loss, and potentiodynamic polarization techniques by Divya et al. (2019). The temperature increased the inhibition efficiency up to 325 K, achieving a maximum inhibition efficiency of about 94.55%, whereas it decreased at higher temperatures (Table 3). According to the PP curves, the *Tithonia diversifolia* flower extract acted as a mixed type inhibitor. Optical electron studies agreed with the strong adsorption inhibitor molecules on the mild steel surface.

Anyiam et al. [140] investigated the sweet potato tuber (PMS) extract, obtained with *n*-hexane as solvent, as corrosion inhibitor on the galvanized steel surface in acidic media (1 M H Cl) at different temperatures and immersion times. Gravimetric and potentiodynamic polarization measurements obtained a maximum inhibition efficiency of 64.26%, obtained at a concentration of 0.7 g/L. Moreover, PP studies showed that the PMS extract behaved as a mixed-type inhibitor. They observed that the corrosion rate increased at all corrosion inhibitor concentrations as temperature increased. In addition, it indicated physical adsorption of the PMS molecules on the galvanized steel according to Gibbs adsorption energy values ($\Delta G^{\circ}_{ads}$) obtained through the Langmuir adsorption isotherm, lower than −20 kJ/mol.

Another extract, tested by Gadow and Motawea, was the one obtained with ginger roots and methanol as solvent [149]. They mentioned that the extract was formed mostly by six organic compounds: gingerol, zingiberene, β-bisabolene, α-farnesene, shogaol, and β-sesquiphellandrene. Through weight loss measurements, the authors obtained an inhibition efficiency of about 92.5% at a 100 ppm concentration (Table 3). The inhibition efficiency increased up to 94% with 200 ppm at 298.15 K. Moreover, the evaluation of the inhibition efficiency at different temperatures showed that the inhibition efficiency decreased as the temperature increased. PP curves indicated that the extract behaved as a mixed-type inhibitor and followed a physical adsorption that was well fitted to a Langmuir isotherm.

Buyuksagis and Dİlek (2019) [144] observed the use of *Papaver somniferum* leaves as a corrosion inhibitor on AISI 304 stainless steel in 0.2 M HCl. The increase in temperature reduced the corrosion inhibition on steel. They attributed this observation to the competition between water molecules and adsorbed inhibitor molecules. A 500 ppm concentration was found to be the best inhibitor, achieving 88% of inhibition efficiency (Table 3). AFM, SEM, and energy-dispersive X-ray spectroscopy (EDS) were used to investigate a metal surface protected with the inhibitor. The metal surface covered with the inhibitor was protected with a thick dense film. The inhibitor was physically adsorbed on the metal surface, as confirmed by the Langmuir isotherm. Moreover, the *Papaver somniferum* leaf extract was found to behave as a mixed-type inhibitor.

Similarly, Emori et al. observed that *Dioscorea septemloba* on carbon steel in a 1 M HCl solution reduced its inhibition properties as temperature increased. Two extracts were obtained, with water and ethanol as solvents. Nuclear magnetic resonance was used to identify the 28 compounds present in ethanol extract. SEM and FT-IR techniques confirmed the adsorption of the extract molecules on the metal surface and the formation of the protective film. The inhibition properties were due to their multiple aromatic rings and heteroatoms found in the compounds. The inhibitor was found as a mixed-type inhibitor according to the EIS tests, achieving 72.1% inhibition efficiency at 2.0 g/L [138] (Table 3).

Furthermore, Ogunleye et al. [142] observed that the inhibition efficiency of *Luffa cylindrica* extract on mild steel in a 0.5 M HCl solution decreases with temperature. In contrast, inhibition efficiency increases as the inhibitor concentration increases. Tanines, flavonoids, phenol, tannins, and alkanol groups are found in the extract by means of gas chromatography–mass spectrometry (GC–MS) and FT-IR tests (Table 3).

The aqueous extract of *Artemisia herba-alba* on stainless steel in 1M $H_3PO_4$ corrosive media was tested by Boudalia and coworkers [137]. They found that the highest inhibition efficiency, of about 88%, was achieved at 1 g/L concentration and 298 K (Table 3). Authors suggested that the decreasing inhibition efficiency as the temperature increased can be understood as a result of the higher dissolution achieved at higher temperatures. The tested inhibitor obeyed the Langmuir adsorption isotherm model. In addition, the calculated Gibbs adsorption energy ($\Delta G°_{ads}$) agreed with a physically adsorbed inhibitor. The protective effect of the *Artemisia herba-alba* aqueous extract on stainless steel was confirmed by means of SEM/EDS micrography.

The performance of *Glycyrrhiza glabra* root extract on the corrosion inhibition on mild steel in 1 M HCl electrolyte was investigated by Alibakhshi et al. [162] through EIS and PP tests. EIS results revealed that the inhibition efficiency increased as the concentration and immersion time increased. A maximum inhibition efficiency of about 88% was achieved at 800 ppm concentration after 24 h immersion (Table 3). Atomic force microscopy confirmed the lower degradation shown on the mild steel treated with licorice extract.

Similarly, Akinbulumo et al. investigated the *Euphorbia heterophylla* Linneo extract on mild steel in a 1.5 M HCl corrosive medium [139]. The gravimetric method was used to measure the inhibition efficiency and corrosion rate. Flory–Huggins adsorption isotherm was a better fit than Langmuir, El-Awary, and Temkin isotherms. Gibbs adsorption energy ($\Delta G°_{ads}$) values below −20 kJ/mol denoted a physisorption process. The maximum efficiency was obtained at 343 K of about 69%, since at higher temperatures the efficiency decreased (Table 3). The authors suggest that this behavior was a consequence of the desorption of the inhibitor molecule from the metal surface [139].

*Ziziphora* leaf extract was proposed as an eco-friendly green inhibitor on mild steel at 1 M HCl concentration by Dehghani and coworkers [150]. Higher concentrations increased the corrosion inhibition efficiency. Moreover, immersion time showed a dependence on the immersion time, obtaining a maximum inhibition efficiency of about 93% with 800 ppm concentration and 2.5 h immersion (Table 3). Gibbs adsorption energy ($\Delta G°_{ads}$) obtained by fitting the Langmuir adsorption isotherm was obtained, ranging from −33 to −35 kJ/mol. Thus, *Ziziphora* leaf extract is a mixed-type physisorption–chemisorption inhibitor.

Haddadi and coworkers, studying the *Junglans regia* green fruit shell extract as a corrosion inhibitor on mild steel in 3.5 wt % NaCl solution, found that the inhibition capacity was promoted with the immersion time up to 48 h [141]. The maximum inhibition efficiency achieved, of about 94%, was obtained with 1000 ppm (Table 3). PP tests showed that both cathodic and anodic reactions were retarded. Functional groups, such as carboxyl, hydroxyl, and carbonyl, of phenolic compounds of the extract were supposed to be physically and chemically adsorbed on the metal surface. Thus, electrostatic interactions and covalent bonding were responsible for a mixed physisorption–chemisorption process [141].

## 2.3. Adsorption Mechanism and Theoretical Characterization

The previous sections exhibited how the most common studies regarding corrosion inhibition were based on experimental evidence obtained through electrochemical tests, used to study the inhibition efficiency and surface microscopy. To a lesser extent, experimental studies were used to determine the way the inhibitor was adsorbed on the metal surface. Gibbs adsorption energy ($\Delta G°_{ads}$), obtained by adjusting a suitable adsorption isotherm, is probably the most reported amount related to the adsorption mechanism. According to its value, $\Delta G°_{ads}$ denotes physisorption, chemisorption, or mixed physisorption–chemisorption. In order gain deeper insight into adsorption mechanisms on

steel surfaces, this subsection focuses on theoretical studies used to complement experimental findings. Theoretical characterization, based on structural analysis and molecule–surface interactions, allows for the elucidation of the adsorption mechanism at an atomic level of detail. Basic information obtained by means of theoretical characterizations of several plant extracts as corrosion inhibitors is summarized in Table 4; the main extract constituents and theoretical framework used for their evaluation are listed as well.

**Table 4.** Theoretical characterization performed for several plant extracts: plant, main extract constituents, and theoretical framework used for their evaluation.

| Plant | Extract Constituents | Theory | Reference |
|---|---|---|---|
| Dioscorea septemloba | Dioscin, β-sitosterol, dioscorone A, and palmitic acid | DFT, MD | [138] |
| Eucalyptus | Macrocarpal E, macrocarpal A, eucalyptome, and ellagic acid | DFT, MC, MD | [127] |
| Eucalyptus globulus | Eucalyptol, globulusin-A, and globulusin-B | DFT | [129] |
| Ficus tikoua | Allantoin, 5-methoxypsoralen, methyl caffeate, and methyl 4-hydroxycinnamate | DFT | [130] |
| Juglans regia | Coumaric acid, ferulic acid, syringic acid, vanillic acid, juglone, and myricetin | DFT, MC, MD | [141] |
| Glycyrrhiza glabra | Licochalcone A, licochalcone E, liquiritigenin, 18β-glycyrrhetinic acid, glycyrrhizin, and glabridin | DFT, MC, MD | [128] |
| Rosa canina | Ascorbic acid, marein, pectin, and tannin | DFT, MC, MD | [146] |
| Saraca ashoka | Epicatechin | DFT | [131] |
| Tamarindus indiaca | Apigenin, naringenin, eriodyctoyl, and taxifolin | DFT, MC, MD | [147] |
| Tamarindus indiaca | Naringenin, apigenin, eriodictyol, and taxifolin | DFT-D | [132] |
| Ziziphora | Acacetin, chrysin, and thymonin | DFT, MC, MD | [150] |

Several characteristics on the compounds contained in a certain extract are known to influence the adsorption and, consequently, the inhibition properties. According to Ben Harb et al. [134], molecular size, carbon chain length, conjugated bonding, aromaticity, aptitude of film to be dense or reticulated, resistance of the bond to the metal substrate, number and nature of bonding groups and atoms within a molecule, and an appropriate solubility of phenolic compounds in solvent extraction are expected to affect the inhibition efficiency. Although the authors obtained an efficient corrosion inhibitor with *Olea europaea* leaf extract, explained by the presence of nitrogen, oxygen heteroatoms, and $\pi$ electrons in the phenol and flavonoid extract content, no more information on the adsorption mechanism was provided.

In contrast, Emori and coworkers reported quantum chemistry studies on the major chemical compounds (dioscin, β-sitosterol, dioscorone A, and palmitic acid) found for *Dioscorea septemloba* extract (Table 4) [138]. Calculations were carried out at the RB3LYP/6-311++G(d,p) level within DFT. Energies of the highest occupied molecular orbital (HOMO) and the lowest unoccupied molecular orbital (LUMO), $E_{HOMO}$ and $E_{LUMO}$, respectively, were used to determine the reactivity of the major components of the extract. They showed, through the HOMO–LUMO energy gap, that the glycoside groups contributed more to the corrosion inhibition performance than the fatty acids. This initial assumption was confirmed by means of molecular dynamics, obtaining binding energies ($E_{bind}$) of the adsorbed molecules on the Fe(110) surfaces following the order of dioscin > dioscorone A > β-sitosterol > palmitic acid. The order found for the number of electrons transferred ($\Delta N$), as defined by Lukovits, also obeyed the same pattern [163]. Lastly, the authors argued that large molecular sizes of the adsorbed molecules ensure greater coverage and improve metal–inhibitor interactions. Moreover, oxygen heteroatoms and $\pi$ electrons can participate by back-bonding from the *d*-orbitals of Fe on the metal surface, establishing covalent bonds.

Dehghani and coworkers studied theoretically the most relevant molecules present in *Eucalyptus* extract: macrocarpal E, macrocarpal A, eucalyptome, and ellagic acid (Table 4). The authors determined

that the $\Delta G^\circ_{ads}$ values within the physisorption–chemisorption regime, ranging from $-32$ to $-35$ kJ/mol, can be directly compared with binding energies calculated by means of theoretical Monte Carlo (MC) and molecular dynamics simulations on Fe(110) surfaces, water, and *Eucalyptus* extract molecules, both neutral and mono-protonated. HOMO, LUMO, and Fukui reactivity indices, obtained through DFT calculations, showed that electron-rich regions (around aromatic rings and double bonds) and oxygen heteroatoms could donate their electrons by electrophilic attack to nucleophiles or empty *d*-orbitals of metal atoms, allowing chemisorption. The physical component is previously described as a consequence of electrostatic and van der Waals interactions [127].

Similarly, Haldhar and Prasad studied, by means of DFT calculations, the three main components of the *Eucalyptus globulus* aqueous extract: eucalyptol, globulusin-A, and globulusin-B (Table 4) [129]. Several global quantum chemical descriptors were used to obtain information regarding the reactivity and the behavior of those compounds in the presence of iron: HOMO–LUMO gap, η softness, σ chemical hardness, and μ dipole moment. Authors suggest that these compounds act as Lewis bases and form coordination bonds with the free *d*-orbital of Fe. HOMOs indicate that inhibitor molecules have pairs of electrons available for nucleophilic interactions, via chemisorption, with low carbon steel surface. Physisorption is possible through the electrostatic interaction among heteroatoms and $Fe^{2+}$ atoms. Back-bonding is possible through π electrons of aromatic rings.

Moreover, Wang and coworkers used the global descriptors, obtained at the B3LYP/6-311++G(d,P) level within DFT, introduced above, to study the corrosion inhibition properties on carbon steel of four major components of the *Ficus tikoua* extract: allantoin, 5-methoxypsoralen, methyl caffeate, and methyl 4-hydroxycinnamate (Table 4) [130]. HOMO and LUMO were used to determine the tendency to donate and to accept electrons of these molecules. Moreover, smaller HOMO–LUMO gap was found to be related to higher corrosion inhibition efficiency. Thus, 5-methoxypsoralen is expected to play the most important role in the inhibition properties of *Ficus tikoua* extract due to its small energy gap. Furthermore, this molecule has a high dipole moment also associated with its high corrosion inhibition performance. Electrostatic potential (ESP) maps were used to describe electrophilic and nucleophilic activities. Nucleophilic regions were mainly distributed near heteroatoms or O-heterocycles, expected to form covalent bond with Fe atoms.

The theoretical study of the aqueous extract of *Juglans regia* was obtained by means of DFT calculations, at the B3LYP/6-311G** level of theory, using Monte Carlo and molecular dynamics [141]. The authors used the optimized configurations of neutral and monoprotonated coumaric acid, ferulic acid, syringic acid, vanillic acid, juglone, and myricetin species (Table 4). All these species were found to be capable of being adsorbed on a Fe(110) surface due to their large negative molecule–surface adsorption energies ($E_{ads}$). Moreover, the configurations observed on the adsorbed molecules showed that aromatic rings and heteroatoms were the most likely to be adsorbed on the metal surface, explained through donor–acceptor interactions [164–166]. HOMOs and LUMOs obtained by DFT calculations confirmed these assumptions, since HOMOs are mostly localized on oxygen heteroatoms and aromatic rings, whereas LUMOs are mostly localized on hydroxyl and carbonyl groups. According to the fraction of electrons transferred ($\Delta N$), neutral molecules tended to donate electrons to the metal surface, whereas monoprotonated species tended to receive charge.

The *Glycyrrhiza glabra* aqueous extract was theoretically studied by DFT calculations at the B3LYP/6-31G** level through six major constituents: licochalcone A, licochalcone E, liquiritigenin, 18β-glycyrrhetinic acid, glycyrrhizin, and glabridin (Table 4) [128]. Moreover, a water solvent was considered due to its self-consistent reaction field (SCRF). Monte Carlo and MD calculations were carried out as well. Aromatic benzene rings, methoxy and carbonyl oxygen centers, and C=C double bonds present in the molecules under study were expected to share their electrons to the empty *d*-orbitals on the iron atoms. Furthermore, species with less-negative HOMO energy were found to give more charges to unfilled *d*-orbitals of surface Fe cations, whereas smaller energy of LUMO denoted the tendency to receive electrons. Small HOMO–LUMO energy gaps found for the selected compounds exhibited their capability to easily share electrons to the metal surface.

Similarly, a theoretical investigation of Sanaei and coworkers regarding the *Rosa canina* aqueous extract was carried out by DFT, at the B3LYP/6-311G*/SCRF level of theory; MC; and molecular dynamics [146]. Four molecules were studied to deepen into the adsorption mechanism of this extract: ascorbic acid, marein, pectin, and tannin (Table 4). Neutral and monoprotonated species were studied. The authors found that the protonation changed HOMO spatially, affecting the active sites responsible for the adsorption on the metallic surface [146]. Conversely, LUMO remained quite similar for neutral and protonated species. The corrosion inhibitor molecules tended to be adsorbed by hydroxyl, carbonyl, and substituted benzene rings. The authors concluded that the corrosion inhibitor compounds were adsorbed by electron transfer interactions.

Saxena et al. studied the *Saraca ashoka* extract as a corrosion inhibitor by modeling the epicatechin molecule through DFT calculations (Table 4) [131]. On the basis of global parameters, such as HOMO, LUMO, and ΔN, the authors suggested that donor–acceptor interactions were established by π electrons of aromatic ring and vacant *d*-orbitals of surface iron atoms. Unshared electron pairs of heteroatoms and vacant *d*-orbitals on iron were possible as well.

Four compounds were chosen by Akbarzadeh and coworkers to study the corrosion inhibition properties of the *Tamarindus indiaca* extract: apigenin, naringenin, eriodyctoyl, and taxifolin (Table 4) [147]. These constituents were modeled by means of DFT, at the B3LYP/6-311G**/Lanl2DZ/SCRF level; MC; and MD approaches. The authors found that Zn–apigenin and Zn–taxifolin complexes tended to be adsorbed by flat orientation on the metal surface. Electron-rich regions, around hydroxyl and 6-membered cycles, and C=C bonds were adsorbed on the iron surface [147].

Similarly, nanocarriers based on a graphene oxide nanoplatform with *Tamarindus indiaca* extract were studied through theoretical approaches [132]. Naringenin, apigenin, eriodictyol, and taxifolin molecules and their zinc(II) cation complexes with GO were studied by dispersion-corrected DFT (DFT-D) (Table 4). The DFT-D level of theory chosen was PBE/DNP. Complexes tended to form π–π interactions and H-bonds among organic molecules and GO layers.

Lastly, the aqueous extract of *Ziziphora* was investigated through DFT, MC, and MD methods. Acacetin, chrysin, and thymonin adopted parallel configurations on the metal surface, leading to a maximized contact area (Table 4). Neutral and monoprotonated species were studied. Electron-rich regions (heterocyclic rings and heteroatoms) were found to be able to donate lone pairs and π electrons to the empty *d*-orbitals on iron atoms. MD simulations in liquid phase obtained binding energies following the order thymonin > acacetin > chrysin. Authors proposed the iron surface charges positively in HCl solution, and thus the chloride ions with negative charge could be adsorbed on the surface. The protonated forms of the *Ziziphora* leaf extract molecules, which carry a net positive electronic charge, can electrostatically interact with adsorbed chloride ions.

## 3. Plant-Based Corrosion Inhibitors in Other Metals

The *Commiphora myrrha* extract, with methanol as extraction solvent, was tested by Al-Nami and Fouda in copper in a 2M $HNO_3$ solution [167]. Weight loss, potentiodynamic polarization, electrochemical frequency (EFM), and electrochemical impedance spectroscopy were used to deepen into the corrosion inhibition properties. WL, EIS, and EFM showed that the inhibition efficiency increased as the inhibitor concentration increased, up to 91.8% at 300 ppm concentration (Table 5). FT-IR, AFM, and SEM techniques were used to confirm the formation of the protective layer formed on the copper surface. PP curves showed that the extract behaves as a mixed-type inhibitor. Caryophyllene and 1,4-methanoazulene were found to be the major inhibitor components of *Commiphora myrrha* extract. The Langmuir adsorption isotherm was found to be the most suitable form.

**Table 5.** Basic information and parameters obtained with several plant extracts evaluated as corrosion inhibitors in copper and aluminum: plant, extraction solvent, metal and corrosive medium used for corrosion inhibition performance tests, temperature and concentration C ranges tested, maximum inhibition efficiency obtained ($\eta_{max}$), and characterizations reported are included.

| Plant | Extraction Solvent | Metal | Corrosion Medium | $\eta_{max}$ (%) | Temp (K) | C | Characterization | Reference |
|---|---|---|---|---|---|---|---|---|
| *Borassus flabellifer* | Water, methanol | Al | 1 M $H_2SO_4$ | 66.8 | 303–333 | 0.1–0.4 g/L | EIS, SEM | [11] |
| *Commiphora myrrha* | Methanol | Cu | 2 M $HNO_3$ | 91 | 298.15–318.15 | 50–300 ppm | EIS, AFM, FT-IR, SEM, PP, WL | [167] |
| *Equisetum arvense* | Methanol | Cu | Seawater | 87.5 | 300 | 250–1000 ppm | EIS, FT-IR, PP, GC–MS | [168] |
| *Hemerocallis fulva* | Methanol | Al | 1 M $H_2SO_4$ | 89 | 303–333 | 200–600 ppm | WL, PP, SEM-EDS, AFM | [169] |

Similarly, the corrosion inhibition properties of *Equisetum arvense* extract on copper substrate in seawater as corrosive medium was studied by Esquivel-Lopez et al. [168]. They found, by means of EIS tests, a maximum inhibition efficiency of about 87.5% (Table 5). Results were confirmed by LPR technique. Gas chromatography coupled to mass spectrometry and FT-IR were used to reveal the chemical structure of the extract constituents: 9,12,15-octadecatrienoic acid, methyl ester, hexadecanoic acid, 2,3-dihydro-2,5-dihydroxy-6-methyl-4H-pyran-4one, sitosterol, campesterol, 5-hydroxymethyl-2-furancarboxaldehyde, hexadecanoic acid, and 9,12-octadecadienoic acid. Morphological characterization, with and without corrosion inhibitor, was obtained by SEM micrographs.

Corrosion inhibition on aluminum, in a 1M $H_2SO_4$ solution, was tested with the *Hemerocallis fulva* extract obtained with methanol as solvent [169]. Results obtained at different concentrations and temperatures exhibit a maximum inhibition efficiency of about 89% at 600 ppm and 303 ± 1 K (Table 5). The adsorption of inhibitor molecules on the metal surface followed the Langmuir isotherm and agreed with physisorption. PP curves agreed with mixed inhibition. P-coumaric acid, ferulic acid, m-coumaric acid, o-coumaric acid, kaempferol, gallic acid, protocatechuic acid, syringic acid, gentisic acid, and quercetin were determined by ultra-high-performance liquid chromatography as the major components of *Hemerocallis fulva* extract. SEM/EDS measurements and AFM were used to study the morphology of the aluminum surface and to confirm the formation of the protective layer.

Moreover, corrosion inhibition in aluminum in a 1M $H_2SO_4$ corrosive medium was studied using water and methanol extracts of *Borassus flabellifer* [11]. PP and EIS revealed that the corrosion inhibition efficiency increased as the inhibitor concentration increased. According to PP measurements, maximum inhibition efficiencies of 66.88% and 51.85% were obtained with methanol and water extracts at 0.40 g/L concentrations, respectively (Table 5). Moreover, the mixed-type inhibition performance of both extracts was determined by PP measurements. SEM micrographs confirmed the formation of a protective film on top of aluminum surface as well as the reduction of the damage caused by corrosion.

All the studies described above highlight an alternative to the current challenges in the industry, for instance, the way in which to obtain novel organic compounds obtained from renewable sources that are being non-toxic and biodegradable [170]. Reports described in this work include natural extracts proven to act as green corrosion inhibitors on different metals. These are obtained from seeds, fruits, leaves, flowers, etc. Notably, corrosion was slowed down, achieving high corrosion inhibition efficiencies up to values around 90% (Table 3). These plant extracts contained phytochemicals, particularly heterocycles, that inhibit the corrosion in an efficient manner. Thus, synthetic organic chemistry groups can be inspired by the compounds compiled here to produce similar corrosion inhibitors. Moreover, large-scale industrial production requires additional processes beyond the parameters shown in this manuscript, opening the door to further experiments and characterizations on plant extracts [170]. Lastly, isolated organic molecules, responsible for the corrosion inhibition effect, must be studied alongside the consideration of other variables, such as inhibitor concentration,

temperature, and release flux. This is all in order to fulfill the norms established for their use in petrochemical, maritime, food, and other industries, such as the reference standard norm (NRF-005-PEMEX-2009) in México [171]. However, one of the drawbacks when evaluating the inhibition efficiency is related to the diversity in chemical composition that present pickles in the field (different salinity). In this way, the use of NACE 1D182 brine is the most suitable, due to its chemical composition, with high content of salts compared to other brines, for example, the one indicated in NACE 1D196 and ASTM D1141.

## 4. Conclusions

The most recent reports on plant extracts that have been evaluated to act as corrosion inhibitors on metal surfaces, mostly in steel, are briefly reviewed. Many variables can be explored to evaluate a plant extract as corrosion inhibitor: concentration, extraction solvent, temperature, and immersion time. The effectiveness of a corrosion inhibitor must be evaluated by at least two electrochemical techniques such as PP, EIS, WL, and others. Constituent compounds of the plant extracts are commonly adsorbed on the metal and are described by the Langmuir model: through physisorption, chemisorption, or mixed mechanisms. Physisorption is usually explained by the interactions among the polar regions of the inhibitor molecules on the metal surface. Chemisorption is due to electron sharing between the inhibitor molecule, from electron-rich regions and heteroatoms, and the metal surface. Phytochemicals obtained from extracts, mostly heterocyclic compounds, are suitable for robust interaction with the metal surface and consequently for the inhibition of corrosion. Theoretical studies, within density functional theory and molecular dynamics theoretical frameworks, are mostly used to elucidate the adsorption mechanism and inhibitor–metal interactions. Lastly, some new contributions on copper and aluminum corrosion inhibition by plant extracts have been discussed as well. Plant extracts obtained corrosion inhibition efficiencies above 60%, most of them around 80–90%. The most important challenge is to have an extract or to isolate the main component that has an inhibition efficiency greater than 90% according to the norm NRF-005-PEMEX- 2009. This compilation can be used as inspiration for research groups to obtain novel organic corrosion inhibitors. Moreover, specific compounds can be isolated and studied, with the aim of producing them in large quantities required for the industry.

**Author Contributions:** A.M. and A.E.V.: Conceptualization, investigation, formal analysis, writing—original draft, writing—review and editing. All authors have read and agreed to the published version of the manuscript.

**Funding:** The APC was funded by Tecnológico de Monterrey through grants for scientific papers publication fund.

**Acknowledgments:** A.M. thanks Tecnológico de Monterrey strategic research group on nanomaterials. Also, A.M. thanks D.S. Tagle-Miralrio for her support with diagrams and figures. Moreover, A.M. thanks J. Muñoz-Villota for her invaluable support during the writing of this manuscript and during the pandemic. A.E.V. and A.M. wish to acknowledge the SNI for the distinction of their membership and the stipend received.

**Conflicts of Interest:** The authors declare no conflict of interest.

## Abbreviations

| | |
|---|---|
| Atomic force microscope | AFM |
| Density functional theory | DFT |
| Dispersion-corrected density functional theory | DFT-D |
| Electrochemical frequency | EFM |
| Electrochemical impedance spectroscopy | EIS |
| Electrostatic potential | ESP |
| Energy-dispersive X-ray spectroscopy | EDS |
| Fourier transform infrared spectroscopy | FT-IR |
| Gas chromatography mass spectrometry | GC–MS |
| Graphene oxide | GO |
| Hydrochloric acid | HCl |

| Highest occupied molecular orbital | HOMO |
| Linear polarization resistance | LPR |
| Lowest unoccupied molecular orbital | LUMO |
| Molecular dynamics | MD |
| Monte Carlo | MC |
| Potentiodynamic polarization | PP |
| Proton-nuclear magnetic resonance | $^1$H NMR |
| Scanning electron microscope | SEM |
| Self-consistent reaction field | SCRF |
| *Tamarindus indiaca* extract | Ti.E/TAM |
| Ultraviolet–visible spectroscopy | UV–VIS |
| Weight loss method | WL |
| X-ray photoelectron spectroscopy | XPS |
| Zinc nitrate | ZS |

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
