# Peer review of "Plant Extracts as Green Corrosion Inhibitors for Different Metal Surfaces and Corrosive Media: A Review"

_processes, doi:10.3390/pr8080942_

Round 1

Reviewer 1 Report

The work is very interesting. The introduction describes topisc exhaustively, but the first part on corrosion and the second one on extraction methods seem unconnected.

Author Response

Comments and Suggestions for Authors

R.- The work is very interesting. The introduction describes topics exhaustively, but the first part on corrosion and the second one on extraction methods seem unconnected.

A.- We thank the reviewer’s comment. The corrosion and extraction methods description were connected by means of introducing the following paragraphs.

Lines 70-80. Although there is a variety of green organic compounds that function as corrosion inhibitors, for example, derivatives of chitosan,[37] phenylmethanimine,[38] and imidazoline,[39] ionic liquid[40] also shows excellent properties protecting the metal surface. These compounds replace the traditional toxic corrosion inhibitors. Highly efficient corrosion inhibitors have been achieved by means of these substances, providing new recycling and reusing routes for drugs as well as corrosion inhibitors obtained from sustainable, ecological and environmentally-friendly sources, being plant extracts a prominent group.[25,41] These extracts constitute another option of great interest, since they offer the possibility of a first approach to determine the class of natural compounds that help inhibit the corrosion process.

Lines 103-108. All plant parts: leaves, flowers, seeds, fruits, roots and stems are used to get the extracts. In brief, extraction methods are based on heating, cooling and separating the active compounds in presence of the solvent.

Traditional extraction methods can be summarized in maceration, infusion decoction, digestion and percolation.[49]

Reviewer 2 Report

The manuscript entitled: “Plant extracts as green corrosion inhibitors for different metal surfaces and corrosive media. A review”, reference: processes-839946-peer-review-v1.

The manuscript is very interesting and impactful. It adequately includes extensive tables that are in my opinion mandatory for Reviews.  It is easy to read, nevertheless it requires extensive improvements, particularly in the formatting of species nomenclature, acronyms and units. The English requires a careful polishing, particularly when the authors use the present tense instead of past tense. The authors should request the help of a native speaker or contract the service of an Editorial Office. A thorough improvement of these issues would in my opinion considerably improve the manuscript quality and readability.

The authors correctly highlight the importance of the different extraction methods, nevertheless, I could not find information stating different or typical types of adsorption, nor the consequences of different approaches. Do the authors do not consider this information important? Please comment.

Furthermore, I consider that the authors should provide a more detailed description of the types of inhibition of corrosion.

I understand that the description of the extraction methods is not the scope of this review, and I acknowledge that the authors nevertheless provided a simplified description, however I would like to provide a suggestion (and it is just a suggestion), in my opinion a Figure displaying the different extraction methodologies in a very simplified manner would be more reader friendly and would not detour the attention from the main issue. Please comment.

Point by point suggestions:

Line 48, why do the authors use “hydrogen chloride” instead of “hydrochloric acid”?

Line 57, please revise the English
Line 62, the authors are repeating the same information from the last sentence. Is it possible to not apply this paragraph and to continue the line of thought from the previous sentence?
Line 92, , " The resultant mixture is separated from the remaining solids by pressing them" I beg your pardon, but I find this sentence unclear.
Between line 89 and line 111 there is not a single reference to support the authors statements. Please revise.
Line 112, I leave one suggestion for the authors: "Solvent has a key role in the extraction methods, since it responsible for solubilizing the active compounds when it diffuses through plant tissues, making their extraction possible" Please feel free to edit it, nevertheless I think that the authors should highlight the importance of solvents in these extractions.
Line 122, is drying always a mandatory step? If so why? This information should be also provided.
Line 126, higher temperature also increases extraction costs correct? In addition it should be "To achieve the optimal concentration"
Line 133, any special reason for several definitions of acronyms to be capitalized, such as: Electrochemical Impedance Spectroscopy? Why not: electrochemical impedance spectroscopy? Please revise throughout the manuscript.

Line 146, in my opinion the term "about" is a colloquial term. Please replace it.
Table 1 is reference in the text after it appears, please revise.

Line 217 and 218, grams and percentage are units, thus they must be separated from their numerical value. Please revise throughout the manuscript.

Line 222, please revise “Leaves extracts also are used” to: Leaves extracts are also used.

Line 242, the authors use “sulphuric acid” and in line 265 its chemical formula. Please define it as acronym and use it consistently.

Line 244, I could not find the definition of Zreal, can the authors please accurately define this acronym?

Line 252, can the authors please provide additional information describing in what consists a mixed-type inhibitor?

Line 272, “Tamarindus indiaca” is a specie, thus its second name can never be capitalized. Please double check all species nomenclature, namely in line 293 and 307.

Line 288 and 321, please revise typos.

Line 313, why do the authors do not use the terms nitrogen and oxygen? Particularly since the term nitrogen (instead of N) is used in line 323.

Line 358, please adequately define the acronym TAM.

Line 383, the authors should be consistent with the temperature units used, meaning, should always use °C or K, not both.

Line 425, please revise typo.

Line 475, “obtaining by fitting someone adsorption isotherm” please revise this colloquial sentence.

Line 563, “pretty similar” please revise colloquial language.

Line 564, revise typo.

Line 642, “Roughly speaking” please revise colloquial language.

Author Response

The manuscript is very interesting and impactful. It adequately includes extensive tables that are in my opinion mandatory for Reviews. It is easy to read, nevertheless it requires extensive improvements, particularly in the formatting of species nomenclature, acronyms and units.

  • We thank the reviewer’s comment. A careful revision of acronyms, species nomenclature and units have been addressed.

The English requires a careful polishing, particularly when the authors use the present tense instead of past tense. The authors should request the help of a native speaker or contract the service of an Editorial Office.

  • We thank the reviewer’s suggestion. The revised version of the manuscript was carefully revised by a native speaker. Correspondent certificate is attached.

A thorough improvement of these issues would in my opinion considerably improve the manuscript quality and readability. The authors correctly highlight the importance of the different extraction methods, nevertheless, I could not find information stating different or typical types of adsorption, nor the consequences of different approaches. Do the authors do not consider this information important? Please comment.

  • We agree with the importance of highlight and describe the adsorption types. More information on the adsorption types was included by the following paragraph:

Lines 211-218. These inhibitors can protect the metal surface in three possible manners: physisorption, chemisorption and film formation. Physisorption is motivated by the electrostatic interaction between inhibitor molecules and the metal surface, whereas chemisorption is due to donor-acceptor interactions between vacant orbitals on the metal surface and free electron pairs in the inhibitor.[78–80] Lastly, film formation provides the metal surface with a physical barrier from the corrosive media, protecting it from corrosive attacks.

Furthermore, I consider that the authors should provide a more detailed description of the types of inhibition of corrosion.

  • We agree with this point as well. The following paragraph was added.

Lines 205-210. Corrosion inhibitors can be classified on the manner they inhibit corrosion: Cathodic, anodic or mixed-type inhibitors. [78–80] Cathodic corrosion inhibitors decrease the corrosion potential, towards lower values, inhibiting the reactions that take place at the cathode, such as oxygen reduction and hydrogen evolution. Anodic corrosion inhibitors move the corrosion potential towards higher values and interacts with the reactive sites on the metal surface, passivating them. Mixed-type inhibitors are those that cannot be classified as cathodic or anodic. [78–80]

I understand that the description of the extraction methods is not the scope of this review, and I acknowledge that the authors nevertheless provided a simplified description, however I would like to provide a suggestion (and it is just a suggestion), in my opinion a Figure displaying the different extraction methodologies in a very simplified manner would be more reader friendly and would not detour the attention from the main issue. Please comment.

  • Figure 2.(Line 157) was included in order to show schemes of traditional extraction methods.

Point by point suggestions:

Line 48, why do the authors use “hydrogen chloride” instead of “hydrochloric acid”?

  • We agree with the reviewer. Hydrogen chloride was replaced.

Lines 56-58. On the other hand, hydrochloric acid (HCl) is frequently used both in decalcification processes and to produce corrosion under controlled conditions.[1] Several authors employ 1M HCl …

Line 57, please revise the English.

  • The following is the revised version the indicated paragraph.

Lines 50-53. To a lesser extent, copper, aluminum and their alloys are studied as well. The high cost associated with corrosion, due to the replacement of rusted metals, can be reduced by using corrosion inhibitors.[10]

Line 62, the authors are repeating the same information from the last sentence. Is it possible to not apply this paragraph and to continue the line of thought from the previous sentence?

  • Line 62 was replaced, and the adjacent paragraphs were merged.

Lines 64-68. Corrosion inhibitors are substances that added in small amounts on metal surfaces or being added to the corrosive medium, reduce their tendency to be affected by corrosion. The use of common corrosion inhibitors is sometimes limited, since these are based on dangerous substances for human health, such as chromium-based treatments.[32]

Line 92, " The resultant mixture is separated from the remaining solids by pressing them" I beg your pardon, but I find this sentence unclear.

  • The referenced sentence was replaced with the following one.

Lines 112-113. The solids suspended in the resulting mixture can be separated by filtering.

Between line 89 and line 111 there is not a single reference to support the authors statements Please revise support the authors statements. Please revise.

  • The paragraph was revised, and the following references were included.

Lines 122-137. Percolation is a filtration method, at room temperature, in which the moistened raw material is placed in a conical vessel, percolator, with an adjustable closure (Figure 2). Then, the percolator must be filled with solvent and covered up, obtaining the extract drop by drop.[50] More sophisticated methods are hot continuous extraction and ultrasound extraction or sonication. The first one uses the Soxhlet apparatus, formed by a glass body with boiling flask, a siphon arm, thimble, extraction chamber and condenser (Figure 2).[51] In brief, the boiling flask containing the solvent is heated and the vapor produced is condensed. The resultant liquid falls into the thimble containing the raw material, and the extract fills the extraction chamber up to put in function the siphon arm to return the liquid into the boiling flask. The reflux process must be stopped up to obtain the degree of extraction desired. Finally, sonication is a technique that uses high energy ultrasounds to improve permeability of cell walls and produces cavitation to disrupt cellular membranes (Figure 2).[52] Consequently, sonication breaks the cells, releasing their content for further extraction. Liquids obtained by the methods introduced above are then clarified by filtration or decantation.

  1. Zhang, Q.-W.; Lin, L.-G.; Ye, W.-C. Techniques for extraction and isolation of natural products: a comprehensive review. Chin. Med. 2018, 13, 20 doi:10.1186/s13020-018-0177-x.
  2. Azmir, J.; Zaidul, I.S.M.; Rahman, M.M.; Sharif, K.M.; Mohamed, A.; Sahena, F.; Jahurul, M.H.A.; Ghafoor, K.; Norulaini, N.A.N.; Omar, A.K.M. Techniques for extraction of bioactive compounds from plant materials: A review. J. Food Eng. 2013, 117, 426–436, doi:10.1016/j.jfoodeng.2013.01.014.
  3. A Review on the Extraction Methods Use in Medicinal Plants, Principle, Strength and Limitation. Med. Aromat. Plants 2015, 04, doi:10.4172/2167-0412.1000196.

Line 112, I leave one suggestion for the authors: "Solvent has a key role in the extraction methods, since it responsible for solubilizing the active compounds when it diffuses through plant tissues, making their extraction possible" Please feel free to edit it, nevertheless I think that the authors should highlight the importance of solvents in these extractions.

  • We took into consideration the reviewer´s comment. The following is the revised version of the paragraph. Some references were included as well.

Lines 140-145. The solvent has a key role in the extraction methods, since it is responsible for solubilizing the active compounds when it diffuses through plant tissues, making their extraction possible. Extraction solvents have shown to affect the physical, chemical and antioxidant properties of the extracts obtained, as the concentration of flavonoids, saponins, phenolic compounds and others present in plant extracts vary according to the extraction solvent.[53–55]

Line 122, is drying always a mandatory step? If so why? This information should be also provided.

  • Since drying is not mandatory, benefits and drawbacks of using fresh plants and dried are discussed. The following is the revised version of the paragraph.

Lines 159-169. Other important parameters for extraction process are drying and extraction temperatures. The first one marks the temperature to get the plant dry, although room temperature is commonly used since plants are usually dried in the shade. The use of dried plants has benefits, being suitable for long storage and with reduced weight. In contrast, the drying process have been shown to affect the stability of bioactive compounds and the antioxidant capacity.[53,59] Fresh plants can be used instead of dried ones, although their use could produce some drawbacks as well. Fresh plants could be subject of degradation by solar radiation, and even some of their constituents could be easily evaporated or oxidized. However, in some cases, phytochemicals could be extracted in higher concentrations from fresh samples instead of dried ones.[60–62]

  1. Pham, H.; Nguyen, V.; Vuong, Q.; Bowyer, M.; Scarlett, C. Effect of Extraction Solvents and Drying Methods on the Physicochemical and Antioxidant Properties of Helicteres hirsuta Lour. Leaves. Technologies 2015, 3, 285–301, doi:10.3390/technologies3040285.

  1. Nóbrega, E.M.; Oliveira, E.L.; Genovese, M.I.; Correia, R.T.P. The Impact of Hot Air Drying on the Physical-Chemical Characteristics, Bioactive Compounds and Antioxidant Activity of Acerola ( M alphigia emarginata ) Residue: Hot Air Dried Acerola Residue. J. Food Process. Preserv. 2015, 39, 131–141, doi:10.1111/jfpp.12213.

Line 126, higher temperature also increases extraction costs correct? In addition it should be "To achieve the optimal concentration"

Lines 172-173. To achieve the optimal concentration of phytochemicals requires the correct choice of extraction temperatures, extraction cost, solvent and others.[63]

Line 133, any special reason for several definitions of acronyms to be capitalized, such as: Electrochemical Impedance Spectroscopy? Why not: electrochemical impedance spectroscopy? Please revise throughout the manuscript.

  • We thank the reviewer’s observation. All acronyms were revised throughout the manuscript.

Line 146, in my opinion the term "about" is a colloquial term. Please replace it.

  • The sentence was revised.

Line 196. Similarly, atomic force microscope (AFM) obtain information regarding the shape of the metal.

Table 1 is reference in the text after it appears, please revise.

  • The location of Table 1 was amended.

Line 217 and 218, grams and percentage are units, thus they must be separated from their numerical value. Please revise throughout the manuscript.

  • Units were revised throughout the manuscript.

Line 222, please revise “Leaves extracts also are used” to: Leaves extracts are also used.

  • That sentence was revised. The following is the revised version.

Line 311. Leaves extracts are also used to obtain extracts with corrosion inhibition properties. For instance,…

Line 242, the authors use “sulphuric acid” and in line 265 its chemical formula. Please define it as acronym and use it consistently.

  • The use of acronyms was revised throughout the manuscript.

Line 331. steel immersed in a 0.5 M sulfuric acid (H2SO4) solution.

Line 244, I could not find the definition of Zreal, can the authors please accurately define this acronym?

  • In order to clarify the term Zreal, the following sentence was included.

Lines 334. This investigation shows that the Nyquist diagram reached a maximum value in the real part of the impedance of about 250 Ωcm2.

Line 252, can the authors please provide additional information describing in what consists a mixed-type inhibitor?

  • The following paragraph with additional information regarding mixed-type inhibitors was added.

Lines 206-207. Corrosion inhibitors can be classified on the manner they inhibit corrosion: Cathodic, anodic or mixed-type inhibitors. [78–80] Cathodic corrosion inhibitors decrease the corrosion potential, towards lower values, inhibiting the reactions that take place at the cathode, such as oxygen reduction and hydrogen evolution. Anodic corrosion inhibitors move the corrosion potential towards higher values and interacts with the reactive sites on the metal surface, passivating them. Mixed-type inhibitors are those that cannot be classified as cathodic or anodic. [78–80]

  1. Brycki, B.E.; Kowalczyk, I.H.; Szulc, A.; Kaczerewska, O.; Pakiet, M. Organic Corrosion Inhibitors. In Corrosion Inhibitors, Principles and Recent Applications; Aliofkhazraei, M., Ed.; InTech, 2018 ISBN 978-953-51-3917-1.
  2. Richardson, J.A. Management of Corrosion in the Petrochemical and Chemical Industries. In Shreir’s Corrosion; Elsevier, 2010; pp. 3207–3229 ISBN 978-0-444-52787-5.
  3. Papavinasam, S. Corrosion Inhibitors. In Uhlig’s Corrosion Handbook; Revie, R.W., Ed.; John Wiley & Sons, Inc.: Hoboken, NJ, USA, 2011; pp. 1021–1032 ISBN 978-0-470-87286-4.

Line 272, “Tamarindus indiaca” is a specie, thus its second name can never be capitalized. Please double check all species nomenclature, namely in line 293 and 307.

  • The use of species names was revised through the manuscript.

Line 288 and 321, please revise typos.

  • The following is the revised version of those sentences.

Lines 387-390. … spectroscopy and potentiodynamic polarization methods are used for deepening into the corrosion inhibitor properties of the extract. The maximum inhibition, of about 70-80 %, is obtained with an 800 ppm concentration (Table 3).

Line 313, why do the authors do not use the terms nitrogen and oxygen? Particularly since the term nitrogen (instead of N) is used in line 323.

  • Terms N and O were replaced with oxygen and nitrogen.

Line 358, please adequately define the acronym TAM.

  • The acronym is introduced with the following sentence.

Line 458. Akbarzadeh et al.[143], studied the performance of green corrosion inhibitor obtained from the Tamarindus indiaca (TAM) extract mixed with zinc nitrate (ZS).

Line 383, the authors should be consistent with the temperature units used, meaning, should always use °C or K, not both.

  • We thank the reviewer’s comment. All temperatures were converted to K.

Line 425, please revise typo.

The revised version of that sentence is the following one.

Line 539. Similarly, Emori, et al., observed that Dioscorea septemloba on carbon steel in a 1 M HCl solution reduces its inhibition properties as temperature increases.

Line 475, “obtaining by fitting someone adsorption isotherm” please revise this colloquial sentence.

  • The following is the revised sentence.

Lines 602-605 Gibbs adsorption energy ΔG°ads, obtained by adjusting a suitable adsorption isotherm, is probably the most reported amount related to the adsorption mechanism.

Line 563, “pretty similar” please revise colloquial language.

  • That phrase was substituted with “quite similar”

Line 564, revise typo.

The following is the revised version of that sentence.

  • Lines 706-708. The authors found that the protonation changes HOMO spatially, affecting the active sites responsible of the adsorption on the   metallic surface.[142]

Line 642, “Roughly speaking” please revise colloquial language.

  • That phrase was replaced.

Line 802. In brief, higher concentration tends to enhance the corrosion inhibition effect,

Reviewer 3 Report

The authors review the literature related to use of plant extracts as corrosion inhibitors. Experimental reports of inhibitor extraction and corrosion inhibition are summarized. Additionally, theoretical evaluations of extraction component adsorption on surfaces are summarized. Most results are for steel corrosion. The literature for copper and aluminum corrosion is limited.

The authors should address the following comments prior to publication:

  1. General comment. The manuscript requires significant editing for grammar and spelling.

  2. General comment. Please provide a Table summarizing nomenclature. For example, WL is not defined in the text.

  3. General comment. The authors should discuss what corrosion inhibitors are used commercially and what inhibition efficiency is required for a natural extract to be competitive.

  4. General comment. The authors should discuss corrosion mechanisms, the characteristics of a mixed-type inhibitor, and how the different types are identified experimentally. Presumably, the authors are referring to differences between physi- and chemi-sorption. This discussion should precede the literature summary and discussion.

  5. Page 1. The abstract does not adequately describe what is reviewed. It is a list of corrosion experimental and theoretical characterization techniques. The abstract should summarize the key findings of the review regarding corrosion inhibition by extracts.

  6. Page 5. The authors claim CR in Table 1 represents the corrosion rate. However, Table 1 contains adsorption isotherms. Isotherms typically are relationships between adsorbent surface coverage and adsorbent concentration in the contacting liquid. Should CR be the adsorbent liquid concentration?

  7. Page 8. The solvent effect section appears to be a sub-section of Section 2 (Plant extracts in steel). Should Section 1.1 be Section 2.1? Similar numbering problems exist for subsequent sub-sections

  8. Page 9. Table 2 should indicate which part of the plant was used to obtain the extract and how the extraction was performed. Additionally, the concentration of the extract in the corrosion tests should be indicated.

  9. Page 13. The authors should state that theoretical studies of extract constituent adsorption are summarized in Table 3 in the second paragraph. The multiple references thereafter to Table 3 can be removed.

  10. Page 16. The section on plant extracts with other metals is numbered Section 2. Is this numbering correct? Please check section and sub-section numbering.

  11. Page 16. Like Table 2, Table 4 should indicate which part of the plant was used to obtain the extract and how the extraction was performed. Additionally, the concentration of the extract in the corrosion tests should be indicated.

  12. Page 17. Most of the inhibition efficiencies in Tables 2 and 3 are between 85 and 95%. In the conclusion please comment on why the efficiencies seem to be narrowly clustered. Additionally, what is the significance of this observation? What is the economic benefit of increasing efficiency from 85 to 95%?

  13. Page 17. Please summarize in the conclusion on what molecular features appear to enhance corrosion inhibition.

Author Response

The authors review the literature related to use of plant extracts as corrosion inhibitors. Experimental reports of inhibitor extraction and corrosion inhibition are summarized. Additionally, theoretical evaluations of extraction component adsorption on surfaces are summarized. Most results are for steel corrosion. The literature for copper and aluminum corrosion is limited.

The authors should address the following comments prior to publication:

  1. General comment. The manuscript requires significant editing for grammar and spelling.
  • We thank the reviewer’s suggestion. The revised version of the manuscript was carefully revised by a native speaker. Correspondent certificate is attached.

  1. General comment. Please provide a Table summarizing nomenclature. For example, WL is not defined in the text.
  • The following abbreviations list was included.

Atomic force microscope    AFM

Density functional theory     DFT

Dispersion-corrected density functional theory         DFT-D

Electrochemical frequency  EFM

electrochemical impedance spectroscopy   EIS

Electrostatic potential         ESP

Energy-dispersive X-ray spectroscopy        EDS

Energy-dispersive X-ray spectroscopy        EDX

Fourier-transform infrared spectroscopy     FT-IR

Gas chromatography mass spectrometry    GC-MS

Graphene oxide     GO

Hydrochloric acid    HCl

Highest occupied molecular orbital  HOMO

Linear polarization resistance          LPR

Lowest unoccupied molecular orbital           LUMO

Molecular dynamics            MD

Monte Carlo           MC

Potentiodynamic polarization          PP

Proton-nuclear magnetic resonance 1H NMR

Scanning electron microscope        SEM

Self-consistent reaction field           SCRF

Tamarindus indiaca extract Ti.E/TAM

  1. General comment. The authors should discuss what corrosion inhibitors are used commercially and what inhibition efficiency is required for a natural extract to be competitive.
    • We compared the inhibition efficiency obtained with plant extracts and that required by the petrochemical industry through the norm

Lines 808-824. All the described above highlights an alternative to face current challenges in the industry. For instance, how to obtain novel organic compounds obtained from renewable sources, being non-toxic and biodegradable.[167] Reports described in this work include natural extracts proved to act as, green, corrosion inhibitors on different metals. These are obtained from seeds, fruits, leaves, flowers, etc. Notably, corrosion has been slow down, achieving high corrosion inhibition efficiencies, up to values around 90% (Table 3). These plant extracts contain phytochemicals, particularly heterocycles, that inhibit the corrosion in an efficient manner. Thus, synthetic organic chemistry groups can be inspired, by the compounds compiled here, to produce similar corrosion inhibitors. Moreover, large-scale industrial production requires additional processes beyond the parameters shown in this manuscript, opening the door to further experiments and characterizations on plant extracts.[167] Lastly, isolated organic molecules, responsible of corrosion inhibition effect, must be studied considering other variables, such as inhibitor concentration, temperature and release flux. All the above in order to fulfill the norms established for their use in petrochemical, maritime, food and other industries; such as the norm NRF-005-PEMEX- 2009.[168] 

Lines 841-845. Since the maximum inhibition, efficiency obtained for the different plant extracts is above 60 %. The most important challenge is to have an extract or isolate the main component that it has an inhibition efficiency greater than 90% according to the norm NRF-005-PEMEX- 2009.

  1. General comment. The authors should discuss corrosion mechanisms, the characteristics of a mixed-type inhibitor, and how the different types are identified experimentally. Presumably, the authors are referring to differences between physi- and chemi-sorption. This discussion should precede the literature summary and discussion.

  • We agree with the importance of highlight and describe the adsorption types. More information on the adsorption types was included by the following paragraph:

Lines 211-218. These inhibitors can protect the metal surface in three possible manners: physisorption, chemisorption and film formation. Physisorption is motivated by the electrostatic interaction between inhibitor molecules and the metal surface, whereas chemisorption is due to donor-acceptor interactions between vacant orbitals on the metal surface and free electron pairs in the inhibitor.[78–80] Lastly, film formation provides the metal surface with a physical barrier from the corrosive media, protecting it from corrosive attacks.

Lines 205-210. Corrosion inhibitors can be classified on the manner they inhibit corrosion: Cathodic, anodic or mixed-type inhibitors. [78–80] Cathodic corrosion inhibitors decrease the corrosion potential, towards lower values, inhibiting the reactions that take place at the cathode, such as oxygen reduction and hydrogen evolution. Anodic corrosion inhibitors move the corrosion potential towards higher values and interacts with the reactive sites on the metal surface, passivating them. Mixed-type inhibitors are those that cannot be classified as cathodic or anodic. [78–80]

  1. Page 1. The abstract does not adequately describe what is reviewed. It is a list of corrosion experimental and theoretical characterization techniques. The abstract should summarize the key findings of the review regarding corrosion inhibition by extracts.
    • We thank the reviewer’s suggestion. The following paragraph was added to the abstract to highlight the main findings, corrosion inhibitions obtained, variables studied and opportunities in this topic.

    • This work reports a broad overview of the current state of the art on the research of new extracts as corrosion inhibitors on metal surfaces in corrosive media. Mostly, constituents obtained from plant extracts are adsorbed on the metal, following the Langmuir adsorption model. Electron rich regions and heteroatoms are found responsible of chemisorption on the metal surface, whereas physisorption is due to the polar regions of the inhibitor molecules. Plant extracts compiled in this work obtained corrosion inhibition efficiencies above 60%, most of them around 80-90%. The effect of concentration, extraction solvent, temperature and immersion time were studied as well. Additional studies regarding plant extracts as corrosion inhibitors on metals are needed to produce solutions for industrial purposes.

  1. Page 5. The authors claim CR in Table 1 represents the corrosion rate. However, Table 1 contains adsorption isotherms. Isotherms typically are relationships between adsorbent surface coverage and adsorbent concentration in the contacting liquid. Should CR be the adsorbent liquid concentration?
    • We thank the reviewer’s observation, CR was incorrectly referred as corrosion rate.The correspondent sentence was corrected with

In Table 2, CR is the concentration of the inhibitor.

  1. Page 8. The solvent effect section appears to be a sub-section of Section 2 (Plant extracts in steel). Should Section 1.1 be Section 2.1? Similar numbering problems exist for subsequent sub-sections
    • Numbering of sections and subsections was revised.

  1. Page 9. Table 2 should indicate which part of the plant was used to obtain the extract and how the extraction was performed. Additionally, the concentration of the extract in the corrosion tests should be indicated.
    • Figure 2 was included to indicate the compounds commonly extracted from the plant parts. Also, Table 3 now includes concentration and temperature ranges tested.

  1.  
  2. Figure 1. Basic parts of a plant and their common active compounds.
  3. Table 3. Basic information and parameters obtained with several plant extracts evaluated as corrosion inhibitors in steel: Plant, extraction solvent, metal and corrosive medium used for corrosion inhibition performance tests, temperature and concentration C test ranges, maximum inhibition efficiency obtained ηmax and characterizations reported are included.

Plant

Extraction

Solvent

Metal

Corrosive medium

Temp

(K)

C

Characterization

ηmax

(%)

Ref

Ananas comosus

Water

Low carbon steel

1 M HCl

308-338

1000 ppm

WL, EIS, PP, UV-Vis, SEM

97.6

[133]

Artemisia herba-alba

Water

Stainless steel

1 M H3PO4

298-353

1 g/L

EIS, SEM-EDS, GC-MS

88

[134]

Bacopa monnieri/

Lawsonia inermis

Water

Low carbon steel

0.5 M NaOH

-

0.4-0.8 g/L

WL, PP

80

[25]

Camellia sinensis

Water

Carbon steel

3.5 % NaCl

-

0.5-2 %

LPR, LC-MS

80

[82]

Cryptocarya nigra

Hexane, dichloromethane, methanol

Mild steel

1 M HCl

-

10-1000 mg/L

PP, EIS, SEM-EDX, FT-IR, UV-Vis

91

[132]

Dioscorea septemloba

Ethanol

Carbon steel

1 M HCl

303.15

0.1- 2.0 g/L

NMR, EIS, FT-IR, SEM

72

[135]

Eucalyptus

Water

Mild steel

1 M HCl

-

800 ppm

PP, WL, EIS, FT-IR, UV-Vis, SEM, AFM

88

[124]

Eucalyptus globulus

Water

Low Carbon Steel

0.5 M H2SO4

-

100-600 mg/L

WL, EIS, AFM, FT-IR, SEM, EFM,

93.09

[126]

Euphorbia heterophylla Llinneo

Water

Mild Steel

1.5 M HCl

-

1 g/L

PP

69

[136]

Ficus tikoua

Water

Carbon steel

1 M HCl

-

200 mg/L

FT-IR, EIS, PP. SEM

95.8

[127]

Glycyrrhiza glabra

Water

Mild Steel

1 M HCl

-

200-600 ppm

EIS, PP, FT-IR, AFM

88

[125]

Ipomoea batatas

N-hexane

Galvanized steel

1 M HCl

308.15-

338.15

0.7g/L

FT-IR, EIS

64.26

[137]

Juglans regia

Water

Mild Steel

3.5 wt% NaCl

-

1000 ppm

PP, EIS, FR-IR, SEM

94.2

[138]

Luffa cylindrica

Ethanol

Mild steel

1 M HCl

303-

333

1 g/L

WL, PP, EIS, FT-IR

87.98

[139]

Nicotiana tabacum

Water

Q235 steel

0.1 M NaOH

-

34 mg/L

PP, FT-IR, XPS, SEM, XRD

83.9

[140]

Olea europaea

Methanol, ethyl acetate, dichloromethane, hexane

Mild steel

0.1 M NaOH + 0.5 M NaCl

-

200-800 mg/L

EIS, GC-MS

91.9

[131]

Papaver somniferum

Ethanol

AISI 304 stainles steel

0.2 M HCl

298.15-

318.15

100-500 ppm

AFM, SEM-EDX

88

[141]

Pterocarpus santalinoides

Water, ethanol, and methanol

Low carbon steel

1 mol/dm3 HCl

298.15- 333.15

0.7 g/L

EIS, PP, LPR, SEM-EDAX, AFM

90

[142]

Rosa canina

Water

Mild steel

1 M HCl

-

200-800 ppm

EIS, UV-Vis, PP, SEM

86

[143]

Saraca ashoka

Water

Mild Steel

0.5 M H2SO4

-

25-100 mg/L

WL, EIS, PP, SEM, UV-Vis, FT-IR, AFM

93.09

[128]

Tamarindus indiaca

Water

Mild steel

3.5 % NaCl

-

300-1000 ppm

FE-SEM, AFM, EIS, FT-IR, GI-XRD

96

[144]

Tamarindus indiaca

Water

Mild steel

3.5 % NaCl

-

1000 ppm

FT-IR, UV–Vis,

XRD, TGA, Raman, EIS, FE-SEM

-

[129]

Taraxacum officinale

Water/ethanol

Carbon steel

Seawater

295.15-

328.15

100-400 mg/L

WL, PP, EIS, SEM, FT-IR, UV-Vis,

94.3

[130]

Tinospora crispa

Water, acetone/water

Mild steel

1 M HCl

-

800-1000 ppm

WL, EIS, SEM

80

[108]

Tithonia diversifolia

Water

Mild steel

1 M HCl

.

0.1-0.7 %

WL, EIS, PP, FT-IR

94.55

[145]

Zingiber officinale

Methanol

Mild Steel

1 M HCl

298.15-

328.15

100 ppm

PP, EIS, FR-IR, UV-Vis, AFM.

92.5

[146]

Ziziphora

Water

Mild steel

1 M HCl

-

200-800 ppm

FT-IR, UV-Vis, EIS

93

[147]

  1. Page 13. The authors should state that theoretical studies of extract constituent adsorption are summarized in Table 3 in the second paragraph. The multiple references thereafter to Table 3 can be removed.
    • Unnecessary references were removed from the text and reference to table 4 is introduce previously.

Basic information, obtained by means of theoretical characterizations, of several plant extracts as corrosion inhibitors is summarized in table Table 4; main extract constituents and theoretical framework used for their evaluation are listed as well.

  1. Page 16. The section on plant extracts with other metals is numbered Section 2. Is this numbering correct? Please check section and sub-section numbering.
    • Numbering of sections and subsections was revised.

  1. Page 16. Like Table 2, Table 4 should indicate which part of the plant was used to obtain the extract and how the extraction was performed. Additionally, the concentration of the extract in the corrosion tests should be indicated.
    • Concentration and temperature ranges tested were included in table 5.

Plant

Extraction Solvent

Metal

Corrosion medium

ηmax

(%)

Temp

(K)

C

Characterization

Ref

Borassus flabellifer

Water, methanol

Al

1 M H2SO4

66.8

303-333

0.1-0.4 g/L

EIS, SEM

[11]

Commiphora myrrha

Methanol

Cu

2 M HNO3

91

298.15-

318.15

50-300 ppm

EIS, AFM, FT-IR, SEM, PP, WL

[164]

Equisetum arvense

Methanol

Cu

Seawater

87.5

300

250-1000 ppm

EIS, FT-IR, PP, GC-MS

[165]

Hemerocallis fulva

Methanol

Al

1 M H2SO4

89

303-333

200-600 ppm

WL, PP, SEM-EDX, AFM

[166]

  1. Page 17. Most of the inhibition efficiencies in Tables 2 and 3 are between 85 and 95%. In the conclusion please comment on why the efficiencies seem to be narrowly clustered. Additionally, what is the significance of this observation? What is the economic benefit of increasing efficiency from 85 to 95%?
    • Corrosion inhibition efficiencies reported in tables are obtained ranging from 64.26 (Ipomoea batatas extract) to 97.6 % (Ananas comosus extract).

    • The impact of highly efficient corrosion inhibitors is highlighted with the following paragraph included in the revised version.

Notably, corrosion has been slow down, achieving high corrosion inhibition efficiencies, up to values around 90% (Table 3). These plant extracts contain phytochemicals, particularly heterocycles, that inhibit the corrosion in an efficient manner. Thus, synthetic organic chemistry groups can be inspired, by the compounds compiled here, to produce similar corrosion inhibitors. Moreover, large-scale industrial production requires additional processes beyond the parameters shown in this manuscript, opening the door to further experiments and characterizations on plant extracts.[167] Lastly, isolated organic molecules, responsible of corrosion inhibition effect, must be studied considering other variables, such as inhibitor concentration, temperature and release flux. All the above in order to fulfill the norms established for their use in petrochemical, maritime, food and other industries; such as the norm NRF-005-PEMEX- 2009.[168] 

  1. Sastri, V.S. Green Corrosion Inhibitors: Theory and Practice; John Wiley and Sons: New Jersey, 2011; ISBN 1-118-01417-0.
  2. Pedraza Basulto, G.K.; Carrillo, I.; Ortega, D.; Martinez, L.; Canto, J. Evaluation at Pipeline Corrosion at Oil Field. ECS Trans. 2015, 64, 103–110, doi:10.1149/06426.0103ecst.

  1. Page 17. Please summarize in the conclusion on what molecular features appear to enhance corrosion inhibition.
    • Molecular properties related to corrosion inhibition and adsorption mechanism are now included in the revised conclusions.

Constituents compounds of the plant extracts are commonly adsorbed and are described by the Langmuir model: through physisorption, chemisorption or mixed mechanisms. Physisorption is usually explained by the interactions among the polar regions of the inhibitor molecules with the metal surface. Chemisorption is due to electron sharing between the inhibitor molecule, from electron rich regions and heteroatoms, and the metal surface. Phytochemicals obtained from extracts, mostly heterocyclic compounds, are suitable to strongly interact with the metal surface and consequently to inhibit corrosion. Theoretical studies are mostly used to elucidate the adsorption mechanism and inhibitor-metal interactions. Density functional theory and molecular dynamics calculations are used for this purpose.

Reviewer 4 Report

Please find comments in the attached file. 

Author Response

Plant extracts as green corrosion inhibitors for different metal surfaces and

corrosion media. A review

This review describes the state of the art of plant extracts as corrosion inhibitors. The authors discussed the effect of solvent type, temperature and immersion time on inhibitor efficiency. The adsorption mechanism and theoretical characterization have been evaluated in the review too. However, the reviewer has major concern as listed below:

Introduction

  • Line 39: “desired state of these metals”
  • We thank the reviewer’s observation.

2) Line 46: hydrogen chloride should be hydrochloric acid

  • The sentence was corrected.

3) Authors should clearly highlight how plant extracts could be considered as

‘green’ corrosion inhibitors.

  • In the revised manuscript, plant extracts as green corrosion inhibitors are highlighted by the following paragraph.

Lines 71-85. Recent approaches take advantage of organic compounds that can be obtained from expired pharmaceutic drugs, mushroom extracts and even plant extracts.[33–36] Although there is a variety of green organic compounds that function as corrosion inhibitors, for example, derivatives of chitosan,[37] phenylmethanimine,[38] and imidazoline,[39] ionic liquid[40] also shows excellent properties protecting the metal surface. These compounds replace the traditional toxic corrosion inhibitors. Highly efficient corrosion inhibitors have been achieved by means of these substances, providing new recycling and reusing routes for drugs as well as corrosion inhibitors obtained from sustainable, ecological and environmentally-friendly sources, being plant extracts a prominent group.[25,41] These extracts constitute another option of great interest, since they offer the possibility of a first approach to determine the class of natural compounds that help inhibit the corrosion process. The advantage is that making an extract from any plant is regarded as an uncomplicated task, thus allowing great possibilities at both extraction and use of these substances for experimentation.

4) Extraction methods could be summarized into a table and explain briefly in text.

  • In order to preserve the consistency through the text, Figure 2 was introduced to synthesize the extraction methods.

Figure 2. Diagrams of common extraction methods. a) Maceration, b) percolation, c) decoction, d) Soxhlet and e) sonication.

5) Introduction could be divided into sections, e.g. Extraction methods,

Characterization, Adsorption and etc.

  • We thank the reviewer’s suggestion; the following subsections were introduced.

1.1. Corrosion inhibition fundamentals

1.2. Extraction methods

1.3. Characterization techniques

1.4. Adsorption mechanism and quantum chemistry methods

6) Authors should highlight the novelty of this review and the different scope

covered as similar review has been published before.

  • We agree with the reviewer. The following paragraph was included in order to highlight the novelty of the topic discussed.

Lines 95-104. According to the literature, extracts of plants, fruits, seeds, flowers and leaves contain active compounds promising for corrosion inhibition in aggressive media. Moreover, these compounds become cheap, widely available and renewable corrosion inhibitor alternatives.[44–48] Thus, a review on the novel plant extracts which have been proved as highly efficient corrosion inhibitors is necessary. This manuscript provides a wide landscape of the recently reported plant extracts as corrosion inhibitors in steel as well as aluminum and copper alloys. Basic aspects of the extraction methods, characterization techniques, theoretical modelling and adsorption mechanisms are discussed as well.

Section 2:

  • Line 210: Persian Liquorice is not in Table 2.

  • We apologize for that error, Persian Liquorice appears as Glycyrrhiza glabra

Lines 304-305. from 64.24 to 98.8 %, reported for Ipomoea batatas extract [123] and Persian liquorice (Glycyrrhiza glabra) extract, [124] …

2) Table 2:

  1. Authors should be consistent in the plant name. If scientific name is used, words such as “ginger” and etc should be changed to scientific name.

  • We thank the reviewer’s observation. All scientific names were clarified and used through the manuscript.

Line 450.

Zingiber officinale

Methanol

Mild Steel

1 M HCl

92.5

PP, EIS, FR-IR, UV-Vis, AFM.

[145]

  1. The concentration of inhibitor used in each study should be included.
  • Concentrations appear in the revised version of Table 3.

  1. If the same plant type is listed, it should be summarized in the same row, e.g. Eucalyptus and Tamarindus indiaca.
  • We thank this suggestion, however, differentiate among plant types is beyond the purpose of this work. Plants properties are listed in alphabetical order to be easily readable.

  • Definition of mixed type inhibitors should be described in detailed.
  • More information regarding mixed type inhibitors was included through the following paragraphs.

Lines 205-210. Corrosion inhibitors can be classified on the manner they inhibit corrosion: Cathodic, anodic or mixed-type inhibitors. [78–80] Cathodic corrosion inhibitors decrease the corrosion potential, towards lower values, inhibiting the reactions that take place at the cathode, such as oxygen reduction and hydrogen evolution. Anodic corrosion inhibitors move the corrosion potential towards higher values and interacts with the reactive sites on the metal surface, passivating them. Mixed-type inhibitors are those that cannot be classified as cathodic or anodic. [78–80]

Lines 359-363. Also, the extract behaves as a mixed-type inhibitor, with both cathodic and anodic reactions. It has been suggested that physically adsorbed molecules (extract) bind to metal in local cathodes and the dissolution of the metal is delayed when the cathodic reaction is prevented, while molecules chemically adsorbed protect the anodic areas.

  • Line 398: please name PMS in full.
    • The abbreviation was included in the following sentence.

Lines 520. Anyiam et al.,[135] investigated the Sweet Potato tubers (PMS) extract, obtained with  n-hexane as solvent.

5) Suggest summarizing the effect of solvent, temperature, and immersion time in

different table for better illustration.

  • Being the most reported parameters, temperature and concentration ranges tested were included in Table 3.

6) Table 3: remark should be added to describe the main findings in each study

  • Table 3 highlights the most important parameters to obtain the maximum corrosion efficiency ηmax, as well as the characterizations done, since it is the most important result reported in each study.

7) Since it is a review paper, authors should provide challenges and opportunities of plant extracts as green corrosive inhibitor. Opinions on the research gaps of the described studies should also be included. How could the potential development in this area could be beneficial in real applications such as petrochemical industry and other sectors should be included as well.

  • We agree with this observation. The following paragraph was introduced to highlight the opportunities to other groups, as well as the challenges

All the described above highlights an alternative to face current challenges in the industry. For instance, how to obtain novel organic compounds obtained from renewable sources, being non-toxic and biodegradable.[167] Reports described in this work include natural extracts proved to act as, green, corrosion inhibitors on different metals. These are obtained from seeds, fruits, leaves, flowers, etc. Notably, corrosion has been slow down, achieving high corrosion inhibition efficiencies, up to values around 90% (Table 3). These plant extracts contain phytochemicals, particularly heterocycles, that inhibit the corrosion in an efficient manner. Thus, synthetic organic chemistry groups can be inspired, by the compounds compiled here, to produce similar corrosion inhibitors. Moreover, large-scale industrial production requires additional processes beyond the parameters shown in this manuscript, opening the door to further experiments and characterizations on plant extracts.[167] Lastly, isolated organic molecules, responsible of corrosion inhibition effect, must be studied considering other variables, such as inhibitor concentration, temperature and release flux. All the above in order to fulfill the norms established for their use in petrochemical, maritime, food and other industries., such as the norm NRF-005-PEMEX- 2009.[168] 

  1. Sastri, V.S. Green Corrosion Inhibitors: Theory and Practice; John Wiley and Sons: New Jersey, 2011; ISBN 1-118-01417-0.

  1. Pedraza Basulto, G.K.; Carrillo, I.; Ortega, D.; Martinez, L.; Canto, J. Evaluation at Pipeline Corrosion at Oil Field. ECS Trans. 2015, 64, 103–110, doi:10.1149/06426.0103ecst.

Quality of presentation

1) There are many abbreviations used in the review, list of abbreviations should

be compiled and included.

  • The following abbreviations list was included.

Atomic force microscope

AFM

Density functional theory

DFT

Dispersion-corrected density functional theory

DFT-D

Electrochemical frequency

EFM

electrochemical impedance spectroscopy

EIS

Electrostatic potential

ESP

Energy-dispersive X-ray spectroscopy

EDS

Energy-dispersive X-ray spectroscopy

EDX

Fourier-transform infrared spectroscopy

FT-IR

Gas chromatography mass spectrometry

GC-MS

Graphene oxide

GO

Hydrochloric acid

HCl

Highest occupied molecular orbital

HOMO

Linear polarization resistance

LPR

Lowest unoccupied molecular orbital

LUMO

Molecular dynamics

MD

Monte Carlo

MC

Potentiodynamic polarization

PP

Proton-nuclear magnetic resonance

1H NMR

Scanning electron microscope

SEM

Self-consistent reaction field

SCRF

Tamarindus indiaca extract

Ti.E/TAM

Ultraviolet–visible spectroscopy

UV-Vis

Weight loss method

WL

X-ray photoelectron spectroscopy

XPS

Zinc nitrate

ZS

Reviewer 5 Report

Dear Authors,

in general, I think that the topic is interesting. However, the paper should be revised and rearranged to be ready for a publication. The introduction and conclusion should be more straightforward and clearer to highlight the importance and the contribution of this review paper.

I have included in the following points in the attachment.

  1. Please check the numbering system.
  2. The paragraphs are not well connected, and there are some typos, missing prepositions, punctuation, and verbs.
  3. There are some inconsistencies in mentioning the techniques, i.e., polarization curves are data representation (line 14), not a technique. It will be better to use a consistent term, for example, potentiodynamic polarization as mostly mentioned by the authors.
  4. Abbreviations should be mentioned before use. There are some abbreviated terms that are not described. For example, ML (Line 294), FT-IT (is this correct? Line 425), WL (Line 596), PL (Table 4), NMR (Line 249, first described in Line 424).
  5. According with --> According to (Line 77, 325, 375)
  6. "Roughly speaking" --> Please change to a more formal transition phrase.
  7. Table 2: "Ginger" is not a scientific name.

Here are some comments and suggestions from me.

  1. The second paragraph can be moved to another place, since the author discusses about metals and mild steel in paragraph 1 and 3. Is this review paper focus only on corrosion in seawater? Please also describe the limitation of the review if necessary.

  1. Line 53: "Among metals, mild steel is the most widely used in oil, food, energy, chemical and construction industries due to its different applications."

--> This sentence is not clear. "Due to its different applications", should be changed to their properties instead. The authors have already mentioned the applications, oil, food, energy, chemical and construction industries. What about the cost, availability, mechanical properties of mild steel?

  1. Line 59: As far as I know, corrosion inhibitor can also be added to the corrosive medium, not only on the metal surface.

  1. Line 65: Examples of green organic inhibitor. What about other sources of green organic inhibitor, e.g., plant oils, biopolymer (chitosan), ionic liquid. What makes plant extract stand out among other options?

  1. Line 70: "An extract is a dissolution composed by the active principles of a plant or its parts (Figure 1) and a certain media acting as solvent".

--> Isn't it supposed to be a solution, instead of a dissolution? I think the sentence should be rephrased.

  1. Line 71: "The extraction yields depend on the solvent used".

--> Does it only depend on the type of solvent? Are there any other factors?

  1. Line 99: If this paragraph intended to explain about more sophisticated methods, then "Percolation" should be described in the previous paragraph.

  1. Line 88-111: It would be more informative to add 1-2 conclusive sentences. Which methods are commonly used, and what are the advantages and disadvantages of using these methods? What is the current development heading towards?

  1. Figure 1: This figure is not so informative. It will be more useful to add some information related to the inhibitors. For example, the chemical structure of organic substances used for inhibitor can be extracted from each part of the plant.

  1. Line 133: EIS has a short testing time compared to which technique? I think LPR can be shorter than EIS as well. Please add a reference.

  1. Line 154: "In order to complement the description on the corrosion inhibition provided by the techniques introduced above, several authors include the study of the adsorption of corrosion inhibitor molecules on the metal surface."

--> What do you mean by description on corrosion inhibition? I think the other characterization method also intend to study the inhibition mechanism.

--> Can you explain more about the findings in the study of adsorption of corrosion inhibitor molecules on the metal surface? Do you mean that there are also a theoretical/simulation studies? If so, please elaborate why it is necessary to complement the experimental study with a simulation.

  1. Table 1: I think this is more suitable for sub section 1.3. Because it is too detail to be presented in the introduction.

  1. Line 175: "Several organic compounds coming from natural sources have been proposed since their compounds exhibit some properties argued as requirements to act as corrosion inhibitors. "

--> Can you provide examples of several organic compounds?

  1. Line 204: "2. Plant extracts in steel"

--> This title is not clear. Please consider changing it. For example, “Plant-based corrosion inhibitors for mild steel”

  1. Line 220: The authors obtained, potentiodynamic polarization, a maximum inhibition efficiency of about 80%.

--> Please rephrase this sentence to make it clearer.

--> Potentiodynamic polarization has been abbreviated before.

  1. Line 270: "The study concludes that Saraca Ashoka seeds aqueous extract behaves as a mixed-type corrosion inhibitor."

--> What is mixed-type corrosion inhibitor? Can you explain more about it?

  1. Page 12: How do most research calculate the inhibition efficiency? The author discusses about the inhibition efficiency based on gravimetry, PP, and EIS. Are the calculation methods the same between these techniques? Which technique is more reliable?

  1. Line 641: Roughly speaking, higher concentration tends to enhance the corrosion inhibition effect.

--> Is there no limitation of the maximum inhibitor concentration? What will happen if the inhibitor is added with a concentration higher than the most efficient concentration?

  1. Conclusion part should be sounder to highlight the most important findings. The authors can also add some future directions of this topic, and the current challenges.

Author Response

Author's Reply to the Review Report (Reviewer 3)

In general, I think that the topic is interesting. However, the paper should be revised and rearranged to be ready for a publication. The introduction and conclusion should be more straightforward and clearer to highlight the importance and the contribution of this review paper. I have included in the following points in the attachment.

  1. Please check the numbering system.
  • Reply: A careful revision was done.
  1. The paragraphs are not well connected, and there are some typos, missing prepositions, punctuation, and verbs.
  • Reply: The manuscript was revised by a native speaker.
  1. There are some inconsistencies in mentioning the techniques, i.e., polarization curves are data representation (line 14), not a technique. It will be better to use a consistent term, for example, potentiodynamic polarization as mostly mentioned by the authors.
  • Reply: We agree with this observation. We changed in the final version as follows:

Lines 12-16. Abstract: Natural extracts have been widely used to protect metal materials from corrosion. The efficiency of these extracts as corrosion inhibitors is commonly evaluated through electrochemical tests, which include techniques such as potentiodynamic polarization, electrochemical impedance spectroscopy and weight loss measures. The inhibition efficiency of different…

  1. Abbreviations should be mentioned before use. There are some abbreviated terms that are not described. For example, ML (Line 294), FT-IT (is this correct? Line 425), WL (Line 596), PL (Table 4), NMR (Line 249, first described in Line 424).
  • Reply: We agree with these observation. We changed in the final version as follows:

Lines 397-400. Deyab and Guibal studied the Taraxacum officinale extract, obtained in a 2:3 aqueous/ethanol solution solvent, on carbon steel in seawater to test desalinization plant conditions. [129] This extract, according to the mass loss method (ML), PP and EIS tests, showed high inhibition efficiencies of up to 94.3 % (Table 3).

Lines 543-544. …ethanol extract. SEM and FT-IR techniques confirmed the adsorption of the extract molecules on the metal surface and the formation of the protective film.

Lines 188-189. Less sophisticated and time-consuming is the weight loss method (WL), since this technique is based on the mass lost by corrosion,…

Lines 341-342. are studied by using FT-IR and proton-nuclear magnetic resonance (1H NMR) techniques, respectively.

Line 769. Commiphora myrrha         Methanol        Copper 2 M HNO3     91        EIS, AFM, FT-IR, SEM, PP, WL      [163]

  1. According with --> According to (Line 77, 325, 375) 6. "Roughly speaking" --> Please change to a more formal transition phrase. 7. Table 2: "Ginger" is not a scientific name
  • Reply: We agree with these observations. We changed as follows with:

Line 436. According to Gibbs adsorption energy ΔG°ads obtained by Langmuir adsorption isotherm plots, the

Zingiber officinale

Methanol

Mild Steel

1 M HCl

92.5

PP, EIS, FR-IR, UV-Vis, AFM.

[145]

Here are some comments and suggestions from me.

  1. The second paragraph can be moved to another place, since the author discusses about metals and mild steel in paragraph 1 and 3. Is this review paper focus only on

corrosion in seawater? Please also describe the limitation of the review if necessary.

  • Reply: we agree and moved the second paragraph:

Lines 53-65. Consequently, solutions to problems related to the degradation by the corrosion of steel, mostly mild steel, is a high-priority topic. To a lesser extent, copper, aluminum and their alloys are studied as well. The high cost associated with corrosion, due to the replacement of rusted metals, can be reduced by using corrosion inhibitors.[10]

Constituting the archetypal source for mineralization and corrosion phenomena, seawater is an example of a common corrosive medium, plenty in chloride ions, such as those present in massive waterbodies around the world: oceans, seas and salt-lakes.[11] On the other hand, hydrochloric acid (HCl) is frequently used both in decalcification processes and to produce corrosion under controlled conditions.[1] Several authors employ 1M HCl solutions as the most prominent corrosion medium to study corrosion, since it is extremely aggressive and can be used to get an idea regarding corrosion on a certain metal.[12–16] Other solutions recurrently used as corrosive media are 3.5 % NaCl,[17–22] 0.1 M HCl[23] and 0.1 M NaOH.[24–26] To a lesser extent, 1M H3PO4 and 0.5 M H2SO4 are used as test solutions as well.[27–31]

Corrosion inhibitors are substances that added in small amounts on metal surfaces or being added to the corrosive medium, reduce their tendency to be affected by corrosion.

  1. Line 53: "Among metals, mild steel is the most widely used in oil, food, energy, chemical and construction industries due to its different applications."

--> This sentence is not clear. "Due to its different applications", should be changed to their properties instead. The authors have already mentioned the applications, oil, food, energy, chemical and construction industries. What about the cost, availability, mechanical properties of mild steel?

  • Reply: We agree with these observations. In the final version, we changed and added the information about the cost.

Lines 44-45. Among metals, mild steel is the most widely used in the oil, food, energy, chemical and construction industries Lines 46-47. This metal shows high mechanical resistance, durability, toughness, among others, which makes it a highly available material and at a relatively low cost.

  1. Line 59: As far as I know, corrosion inhibitor can also be added to the corrosive medium, not only on the metal surface.

  • Reply: We completed this sentence with your observation.

Lines 64-68. Corrosion inhibitors are substances that added in small amounts on metal surfaces or being added to the corrosive medium, reduce their tendency to be affected by corrosion. The use of common corrosion inhibitors is sometimes limited, since these are based on dangerous substances for human health, such as chromium-based treatments.[32]

  1. Line 65: Examples of green organic inhibitor. What about other sources of green organic inhibitor, e.g., plant oils, biopolymer (chitosan), ionic liquid. What makes plant extract stand out among other options?

  • Reply: We added the examples of green organic inhibitor as follows:

Lines 68-80. Recent approaches take advantage of organic compounds that can be obtained from expired pharmaceutic drugs, mushroom extracts and even plant extracts.[33–36] Although there is a variety of green organic compounds that function as corrosion inhibitors, for example, derivatives of chitosan,[37] phenylmethanimine,[38] and imidazoline,[39] ionic liquid[40] also shows excellent properties protecting the metal surface. These compounds replace the traditional toxic corrosion inhibitors. Highly efficient corrosion inhibitors have been achieved by means of these substances, providing new recycling and reusing routes for drugs as well as corrosion inhibitors obtained from sustainable, ecological and environmentally-friendly sources, being plant extracts a prominent group.[25,41] These extracts constitute another option of great interest, since they offer the possibility of a first approach to determine the class of natural compounds that help inhibit the corrosion process. The advantage is that making an extract from any plant

  1. Zhao, Q.; Guo, J.; Cui, G.; Han, T.; Wu, Y. Chitosan derivatives as green corrosion inhibitors for P110 steel in a carbon dioxide environment. Colloids Surf. B Biointerfaces 2020, 194, 111150, doi:10.1016/j.colsurfb.2020.111150.
  2. Machado Fernandes, C.; Pina, V.G.S.S.; Alvarez, L.X.; de Albuquerque, A.C.F.; dos Santos Júnior, F.M.; Barrios, A.M.; Velasco, J.A.C.; Ponzio, E.A. Use of a theoretical prediction method and quantum chemical calculations for the design, synthesis and experimental evaluation of three green corrosion inhibitors for mild steel. Colloids Surf. Physicochem. Eng. Asp. 2020, 599, 124857, doi:10.1016/j.colsurfa.2020.124857.
  3. Munis, A.; Zhao, T.; Zheng, M.; Rehman, A.U.; Wang, F. A newly synthesized green corrosion inhibitor imidazoline derivative for carbon steel in 7.5% NH4Cl solution. Sustain. Chem. Pharm. 2020, 16, 100258, doi:10.1016/j.scp.2020.100258.
  4. Verma, C.; Ebenso, E.E.; Quraishi, M.A. Ionic liquids as green and sustainable corrosion inhibitors for metals and alloys: An overview. J. Mol. Liq. 2017, 233, 403–414, doi:10.1016/j.molliq.2017.02.111.

  1. Line 70: "An extract is a dissolution composed by the active principles of a plant or its parts (Figure 1) and a certain media acting as solvent".

--> Isn't it supposed to be a solution, instead of a dissolution? I think the sentence should be rephrased.

  • Reply: We changed in the final version as follows:

Lines 83-84. An extract is a solution composed by the active principles of a plant or its parts (Figure 1) and a certain medium acting as solvent

  1. Line 71: "The extraction yields depend on the solvent used". --> Does it only depend on the type of solvent? Are there any other factors?

  • Reply: We agree with this observation.

Line 84. The extraction yields depend on the polarity of the solvent used, in the techniques or methods (Soxhlet and maceration), among others.

  1. Line 99: If this paragraph intended to explain about more sophisticated methods, then "Percolation" should be described in the previous paragraph.

  • Reply: We agree with this observation. Percolation was introduced previously.

  1. Line 88-111: It would be more informative to add 1-2 conclusive sentences. Which methods are commonly used, and what are the advantages and disadvantages of using

these methods? What is the current development heading towards?

  • Reply: We added more information about these techniques as follows:

Lines 113-115. For this method, the advantage is that all the essence is extracted without altering it in the least, and the active ingredients are easily soluble.

Lines 137-139. The advantages of percolation lie in the high performance of active substances, in the short time required for their manufacture and the economy of the materials used.

  1. Figure 1: This figure is not so informative. It will be more useful to add some information related to the inhibitors. For example, the chemical structure of organic substances used for inhibitor can be extracted from each part of the plant.

  • Reply: In the figure 1, we added the organic compounds found in parts of a plant.

Figure 1. Basic parts of a plant and their common active compounds.

  1. Line 133: EIS has a short testing time compared to which technique? I think LPR can be shorter than EIS as well. Please add a reference.

  • Reply: Is correct to LPR is shorter than EIS, But EIS shows more information for example: mechanism, different resistance of the system.

  1. Vaamonde, A.J.V.; de Damborenea, J.J.; González, J.J.D. Ciencia e ingeniería de la superficie de los materiales metálicos; Editorial CSIC-CSIC Press, 2000; Vol. 31; ISBN 84-00-07920-5.

  1. Line 154: "In order to complement the description on the corrosion inhibition provided by the techniques introduced above, several authors include the study of the adsorption of corrosion inhibitor molecules on the metal surface."

--> What do you mean by description on corrosion inhibition? I think the other characterization method also intend to study the inhibition mechanism.

  • Reply: we refer to the type of adsorption (physical or chemical) that the molecules carry out on the metal surface

Lines 268-270. In order to complement the description on the corrosion inhibition provided by the techniques introduced above, several authors include the study of the adsorption of corrosion inhibitor molecules on the metal surface.[95,103–105]

--> Can you explain more about the findings in the study of adsorption of corrosion inhibitor molecules on the metal surface? Do you mean that there are also a theoretical/simulation studies? If so, please elaborate why it is necessary to complement the experimental study with a simulation.

  • Reply: We have included the subsection 2.3. “Adsorption mechanism and theoretical characterization” belonging theoretical studies and adsorption mechanisms regarding organic corrosion inhibitors on steel surfaces. The first paragraph introduces the importance of the theoretical characterization and analysis of adsorption mechanism.

Lines 598-609. Previous sections exhibit how the most common studies regarding corrosion inhibition are based on experimental evidence obtained through electrochemical tests, used to study the inhibition efficiency and surface microscopy. To a lesser extent, experimental studies are used to determine the way the inhibitor is adsorbed on the metal surface. Gibbs adsorption energy ΔG°ads, obtained by adjusting a suitable adsorption isotherm, is probably the most reported amount related to the adsorption mechanism. According to its value, ΔG°ads denotes physisorption, chemisorption or mixed physisorption-chemisorption. In order to deepen into adsorption mechanisms on steel surfaces, this subsection focuses on theoretical studies used to complement experimental findings. Theoretical characterization, based on structural analysis and molecule-surface interactions, allows to elucidate the adsorption mechanism at an atomic level of detail.

  1. Table 1: I think this is more suitable for sub section 1.3. Because it is too detail to be presented in the introduction.

  • Reply: We agree with this observation. The table 1 was changed to number and section 2 as follows:

Lines 268-288. In order to complement the description on the corrosion inhibition provided by the techniques introduced above, several authors include the study of the adsorption of corrosion inhibitor molecules on the metal surface.[95,103–105] Several adsorption isotherm models have been proposed to describe the adsorption mechanism of organic inhibitors on the metal surface. The most appropriated model is the one that best fits to the experimental values. The following table (Table 2) summarizes the most used adsorption isotherm models. In Table 2, CR is the corrosion rate, θ the degree or surface coverage of inhibitor, Kads adsorption equilibrium constant. In most cases,[2,106–109] the model that best fits to extracts as corrosion inhibitors on metals is the Langmuir isotherm. Adsorption equilibrium constant Kads is directly related to Gibbs adsorption energy ΔG°ads by the equation:

                         (1)

Where R is the universal gas constant, T is the absolute temperature, Kads is the adsorption equilibrium constant and 55.5 is the water solution concentration in mol/L units. The negative sign of ΔG°ads denotes the spontaneous adsorption of the corrosion inhibitor molecules on the metal surface. Values below -40 kJ/mol are related to chemisorption, ranging from -40 to -20 kJ/mol to mixed physisorption-chemisorption regime and above -20 kJ/mol to physisorption.[110–116]

Table 2. Main adsorption isotherm models.

Model

Adsorption isotherm

Ref

Langmuir

[117,118]

Temkin

[117,119,120]

Freundlish

[120,121]

Flory-Huggins

[121]

Frumkin

[117]

El-Awady

[117,122]

  1. Line 175: "Several organic compounds coming from natural sources have been proposed since their compounds exhibit some properties argued as requirements to act as corrosion inhibitors. "

--> Can you provide examples of several organic compounds?

·         Reply:  Within the plant extracts studied, it has been found that (-)-epigallocatechin gallate, (-)-epicatechin gallate, catechin or (-)-epicatechin and vallinin, all present in vanilla.

Lines 222-224. Several organic compounds coming from natural sources, such as (-)-epicatechin gallate, catechin or (-)-epicatechin and vallinin, have been proposed since their compounds exhibit some properties argued as requirements to act as corrosion inhibitors.

  1. Line 204: "2. Plant extracts in steel"

--> This title is not clear. Please consider changing it. For example, “Plant-based corrosion inhibitors for mild steel”

  • Reply: we agree with suggestion and changed for:

Line 257. 2. Plant-based corrosion inhibitors for mild steel

  1. Line 220: The authors obtained, potentiodynamic polarization, a maximum inhibition efficiency of about 80%.

--> Please rephrase this sentence to make it clearer.

--> Potentiodynamic polarization has been abbreviated before.

  • Reply: We rewrite the sentence in the final version as follows:

Lines 304-306. The authors found a maximum inhibition efficiency about 80% by the PP technique. Besides, weight loss measurements resulted in an inhibition efficiency of 65 % (Table 3).

  1. Line 270: "The study concludes that Saraca Ashoka seeds aqueous extract behaves as a mixed-type corrosion inhibitor."

--> What is mixed-type corrosion inhibitor? Can you explain more about it?

  • Reply: The mixed type corrosion inhibitor it means to the inhibitor extract being adsorbed on the metal surface. This observation causes that the dissolution of the metal was retarded while molecules chemically adsorbed protect the anodic In the final version, we added this information.

Lines 359-363. Also, the extract behaves as a mixed-type inhibitor, with both cathodic and anodic reactions. It has been suggested that physically adsorbed molecules (extract) bind to metal in local cathodes and the dissolution of the metal is delayed when the cathodic reaction is prevented, while molecules chemically adsorbed protect the anodic areas.

  1. Page 12: How do most research calculate the inhibition efficiency? The author discusses about the inhibition efficiency based on gravimetry, PP, and EIS. Are the

calculation methods the same between these techniques?

Which technique is more reliable?

  • Reply: Each of the techniques provides us with certain information, so they are complementary. In the final version, we added the equations to determine inhibition efficiency by different techniques.

Lines 261-268. Table 1 shows the equation (1-3) to determine the inhibition efficiency (IE) by different techniques: Table 1. Equation to determine the inhibition efficiency

Technique

Equation

EIS

IE (%) = (1- ((Rp) blank / (Rp) inhibitor) * 100             (1)

PP

                  (2)

WL

IE (%) = ((W0-W1) / (W0)) * 100              (3)

Where Rp is a polarization resistance, with inhibitor, it is (Rp) inhibitor, and without it, it is (Rp) blank, icorr is the current density with and without inhibitor. W0 and Wi are the weight loss values in absence and presence of the inhibitor.

  1. Line 641: Roughly speaking, higher concentration tends to enhance the corrosion inhibition effect.

--> Is there no limitation of the maximum inhibitor concentration?

What will happen if the inhibitor is added with a concentration higher than the most efficient concentration?

 ·         Reply: According to the norm NRF-005-PEMEX- 2009 the maximum concentration to determine that it is a good inhibitor should be used a maximum of 50 ppm. However, it is not a rule that increasing the concentration increases the inhibition efficiency. 

Line 804. In brief, higher concentration tends to enhance the corrosion inhibition effect, but it is…

  1. Conclusion part should be sounder to highlight the most important findings. The authors can also add some future directions of this topic, and the current challenges

  • Conclusions was reformulated as follows.

The most recent reports on plant extracts evaluated to act as corrosion inhibitors on metal surfaces, mostly in steel, are briefly reviewed. Many variables can be explored to evaluate a plant extract as corrosion inhibitor: concentration, extraction solvent, temperature and immersion time. The effectiveness of a corrosion inhibitor must be evaluated by at least two electrochemical techniques such as PP, EIS, WL, others. Constituents compounds of the plant extracts are commonly adsorbed and are described by the Langmuir model: through physisorption, chemisorption or mixed mechanisms. Physisorption is usually explained by the interactions among the polar regions of the inhibitor molecules with the metal surface. Chemisorption is due to electron sharing between the inhibitor molecule, from electron rich regions and heteroatoms, and the metal surface. Theoretical studies are mostly used to elucidate the adsorption mechanism and inhibitor-metal interactions. Density functional theory and molecular dynamics calculations are used for this purpose. Lastly, some new contributions on copper and aluminum corrosion inhibition by plant extracts are discussed as well. Since the maximum inhibition, efficiency obtained for the different plant extracts is above 60 %. The most important challenge is to have an extract or isolate the main component that it has an inhibition efficiency greater than 90% according to the norm NRF-005-PEMEX- 2009. This compilation can be used as inspiration for synthetic organic groups to obtain novel organic corrosion inhibitors. Also, specific compounds can be isolated and studied, with the aim of produce them in the large quantities required for the industry.

Round 2

Reviewer 2 Report

The manuscript entitled: “Plant extratcts as green corrosion inhibitos for different metal surfaces and corrosive media. A review”, reference 839946-V2.

I would like to congratulate the authors for the extensive improvements of the manuscript. In my sincere opinion, its quality and potential impact have been considerably improved.

Please just revise the following: Line 744 and 758 (¡Error! No se encuentra el origen de la referencia.)

Author Response

I would like to congratulate the authors for the extensive improvements of the manuscript. In my sincere opinion, its quality and potential impact have been considerably improved.

Please just revise the following: Line 744 and 758 (¡Error! No se encuentra el origen de la referencia.)

Reply: We thank the valuable reviewer’s comments. Both references were amended.

Reviewer 3 Report

The authors attempted to address reviewer comments.

Author Response

We thank the valuable reviewer’s comments.

Reviewer 4 Report

Introduction

  • Section 1.2: Please indicate clearly Figure 2 (a)/(b) or etc in the text.
  • Line 126: remove ‘in the least’.
  • Line 177: ‘subjected to’
  • Line 197: system

Section 2:

  • Equation 1: ‘ads’ should be subscripted
  • Line 330 – 331: check reference
  • Line 512 – 513: check reference
  • Line 630: delete one of the ‘table’
  • Line 744: check reference
  • Line 758: check reference
  • List of abbreviations: please be consistent on EDS and EDX. Only one term should be used.

Author Response

Section 1.2: Please indicate clearly Figure 2 (a)/(b) or etc in the text.

Line 126: remove ‘in the least’.

Line 177: ‘subjected to’

Line 197: system

  • Reply: All suggestions were considered.

Section 2:

Equation 1: ‘ads’ should be subscripted

Line 330 – 331: check reference

Line 512 – 513: check reference

Line 630: delete one of the ‘table’

Line 744: check reference

Line 758: check reference

  • Reply We thank the valuable reviewer’s corrections. All references were amended and corrections were done.

List of abbreviations: please be consistent on EDS and EDX. Only one term should be used.

  • EDS was assumed as the abbreviation throughout the manuscript.

Reviewer 5 Report

Dear Authors,

Thank you for the explanation and for carefully editing the manuscript.

The authors have put a lot of effort in developing it. In my opinion, there is a significant improvement in the clarity and soundness of the manuscript.

However, there are some new comments/suggestions based on the revised version that I think will further improve the quality and readability of the manuscript. Please refer to the details attached for minor errors, responses from the previous round, and new suggestions. I hope it will be helpful.

Author Response

Dear Authors,

Thank you for the explanation and for carefully editing the manuscript.

The authors have put a lot of effort in developing it. In my opinion, there is a significant improvement in the clarity and soundness of the manuscript.

However, there are minor errors and some new comments/suggestions based on the revised version that I think will further improve the quality and readability of the manuscript. Please refer to the details attached for minor errors, responses from the previous round, and new suggestions. I hope it will be helpful.

Minor errors

Line 16: ” measures”, shouldn’t this be measurement?

Line 197: system (typo)

Line 200: abbreviate LPR

Line 220: “Cathodic”, why does it start with a capital letter?

Line 330, 512, 744, 758: Please check again the error in referencing.

Line 502: “PP studies According..”, I think there is an error in punctuation, or the sentence is incomplete.

Line 588, 609: “24 h”, shouldn’t this be “hours”?

Line 841: Suggestion, ”adsorbed on the metal”

  • Reply: All these minor corrections were addressed.

Line 848-850: Is it possible to merge these sentences?

  • Both sentences were merged as follows:

Lines 853-855. Theoretical studies, within density functional theory and molecular dynamics theoretical frameworks, are mostly used to elucidate the adsorption mechanism and inhibitor-metal interactions.

Some follow ups

Point 4: Line 65: Examples of green organic inhibitor. What about other sources of green organic inhibitor, e.g., plant oils, biopolymer (chitosan), ionic liquid. What makes plant extract stand out among other options?

  • Reply: We added the examples of green organic inhibitor as follows: line 76-88

Follow up:

I think the authors have added some good references to the manuscript. The authors have also added the advantage of using plant extract -based corrosion inhibitor. However, there is an incomplete sentence that needs to be revised (line 76-79). In addition, I would like to suggest using a more exact term for “great possibilities” (line 86-88), for example: faster, more efficient, or cheaper? (if possible).

Line 76-79: Although there is a variety of green organic compounds that function as corrosion inhibitors, for example, derivatives of chitosan,[37] phenylmethanimine,[38] and imidazoline,[39] ionic liquid[40] also shows excellent properties protecting the metal surface.

  • Reply: We changed the sentences as follows:

……There is a variety of green organic compounds that function as corrosion inhibitors and shows excellent properties protecting the metal surface for example, derivatives of chitosan,[38] phenylmethanimine,[39] and imidazoline,[40] ionic liquid[41]. In consequence……..

Line 86-88: The advantage is that making an extract from any plant is regarded as an uncomplicated task, thus allowing great possibilities at both extraction and use of these substances for experimentation.

  • Reply: We agree with observation and changed in the sentence:

…… The advantage is that making an extract from any plant is regarded as an uncomplicated task, thus allowing more efficient at both extraction and use of these substances for experimentation.

Point 8: Line 88-111: It would be more informative to add 1-2 conclusive sentences. Which methods are commonly used, and what are the advantages and disadvantages of using these methods? What is the current development heading towards?

  • Reply: We added more information about these techniques as follows:

Lines 113-115. For this method, the advantage is that all the essence is extracted without altering it in the least, and the active ingredients are easily soluble.

Lines 137-139. The advantages of percolation lie in the high performance of active substances, in the short time required for their manufacture and the economy of the materials used.

Follow up:

This is a good addition to the manuscript. However, I think the sentence below (line 149) should be added after the description of percolation technique (line 136) instead of at the end of the next paragraph.

Line 149: The advantages of percolation lie in the high performance of active substances, in the short time required for their manufacture and the economy of the materials used.

  • Reply: We changed the paragraph in line 136 as follows:

….. percolator, with an adjustable closure (Figure 2b). Then, the percolator must be filled with solvent and covered up, obtaining the extract drop by drop.[51] The advantages of percolation lie in the high performance of active substances, in the short time required for their manufacture and the economy of the materials used.

More sophisticated methods are hot continuous extraction and ultrasound extraction or sonication.[52] The.........

Point 10: Line 133: EIS has a short testing time compared to which technique? I think LPR can be shorter than EIS as well. Please add a reference.

  • Reply: Is correct to LPR is shorter than EIS, But EIS shows more information for example: mechanism, different resistance of the system.

Follow up:

Thank you for the explanation and the added reference. In my opinion, with the revised sentence, it is better to remove the part “compared with LPR and PP technique”, because this contradicts the explanation that LPR takes shorter time than EIS. The authors can also add the information obtained from EIS measurements, as mentioned on the reply above.

Line 193: Another recurrently used technique to study anti-corrosion performance, in considerable short testing times, is electrochemical impedance spectroscopy (EIS) compared with linear polarization resistance (LPR) or the PP technique.

  • Reply: We agree and changed this sentence as follows:

…. Another recurrently used technique to study anti-corrosion performance, in considerable short testing times, is electrochemical impedance spectroscopy (EIS).[66].....

……. This technique is used to determine the impedance of a system in terms of the frequency of a variable potential. The analysis of EIS results relies on models with equivalent electrical circuits and the most recurrent graphical representations of its results are Nyquist plots. [67] EIS shows more information for example: mechanism, different resistance of the system………

Point 17: Page 12: How do most research calculate the inhibition efficiency? The author discusses about the inhibition efficiency based on gravimetry, PP, and EIS. Are the

calculation methods the same between these techniques?

Which technique is more reliable?

  • Reply: Each of the techniques provides us with certain information, so they are complementary. In the final version, we added the equations to determine inhibition efficiency by different techniques.

Lines 261-268. Table 1 shows the equation (1-3) to determine the inhibition efficiency (IE) by different techniques: Table 1. Equation to determine the inhibition efficiency

Technique

Equation

EIS

IE (%) = (1- ((Rp) blank / (Rp) inhibitor) * 100             (1)

PP

                  (2)

WL

IE (%) = ((W0-W1) / (W0)) * 100              (3)

Where Rp is a polarization resistance, with inhibitor, it is (Rp) inhibitor, and without it, it is (Rp) blank, icorr is the current density with and without inhibitor. W0 and Wi are the weight loss values in absence and presence of the inhibitor.

Follow up:

I think Table 1 is a very informative addition to the manuscript. Please add references to the equations.

  • Reply: We added the references in table 1

            Aldana-González, J.; Espinoza-Vázquez, A.; Romero-Romo, M.; Uruchurtu-Chavarin, J.; Palomar-Pardavé, M. Electrochemical evaluation of cephalothin as corrosion inhibitor for API 5L X52 steel immersed in an acid medium. Arab. J. Chem. 2019, 12, 3244–3253, doi:10.1016/j.arabjc.2015.08.033.

  1. Saxena, A.; Sharma, V.; Thakur, K.K.; Bhardwaj, N. Electrochemical Studies and the Surface Examination of Low Carbon Steel by Applying the Extract of Citrus sinensis. J. Bio- Tribo-Corros. 2020, 6, 41, doi:10.1007/s40735-020-00338-x.
  2. Abod, B.M.; Al-Alawy, R.M.; Khadom, A.A.; Kamar, F.H. Experimental and Theoretical Studies for Tobacco Leaf Extract as an Eco-friendly Inhibitor for Steel in Saline Water. J. Bio- Tribo-Corros. 2019, 5, 75, doi:10.1007/s40735-019-0268-y.

Additional comments and suggestions

  1. Line 852: Since the maximum inhibition, efficiency obtained for the different plant extracts is above 60 %.

à I think this sentence is incomplete, please rephrase or develop this sentence.

  • The sentence was reformulated as follows:

Lines 853. Plant extracts obtained corrosion inhibition efficiencies above 60%, most of them around 80-90%.

  1. Line 855: This compilation can be used as inspiration for synthetic organic groups to obtain novel organic corrosion inhibitors.

à “Synthetic organic groups here”, does it refer to research groups?

  • The sentence as corrected as follows:

This compilation can be used as inspiration for research groups to obtain novel organic corrosion inhibitors.

  1. Line 831: Is there any equivalent international standard of norm NRF-005-PEMEX- 2009? If not, please mention what is the norm about and from which country/organization it is based on.

  • Reply: We added the information of Norm

……..; such as the reference standard norm (NRF-005-PEMEX- 2009) in México .[168]  However, one of the drawbacks when evaluating the inhibition efficiency is related to the diversity in chemical composition that present pickles in the field (different salinity). In this way, the use of NACE 1D182 brine is the most suitable, due to its chemical composition with high content of salts compared to other brines, for example, the one indicated in NACE 1D196 and ASTM D1141……..
